# Single-Step Bidirectional Unpaired Image Translation Using Implicit Bridge Consistency Distillation

## Abstract

Unpaired image-to-image translation has seen significant progress since the introduction of CycleGAN. However, methods based on diffusion models or Schrödinger bridges have yet to be widely adopted in real-world applications due to their iterative sampling nature. To address this challenge, we propose a novel framework, Implicit Bridge Consistency Distillation (IBCD), which enables single-step bidirectional unpaired translation without using adversarial loss. IBCD reformulates consistency distillation by using a diffusion implicit bridge model that connects PF-ODE trajectories between distributions, with a novel design parametrization to enable effective translation in a single step. Additionally, we introduce two key improvements: 1) distribution matching for consistency distillation and 2) adaptive weighting method based on distillation difficulty. Experimental results demonstrate that IBCD achieves state-of-the-art performance on benchmark datasets in a single generation step.

## 1 Introduction

Unpaired image-to-image (I2I) translation (Zhu et al., 2017a), which transfers images between domains while preserving content without supervision, has gained continuous attention in academia and industry. This approach is particularly useful in real-world scenarios where paired data is hard to obtain, such as in medical and scientific imaging (Kaji & Kida, 2019; Chen et al., 2023). Despite the recent advances in modern zero-shot image editing methods (Parmar et al., 2023; Hertz et al., 2023; 2022), their applicability remains limited due to challenges such as the lack of domain-specific adaptation and the difficulty of preserving fine-grained details. Even the latest foundational models, such as GPT-Image-1, consistently underperform without domain-specific adaptation (Tab. 2). Therefore, unpaired I2I translation remains essential for applications like image enhancement, artifact removal, and cross-modality translation in modern computer vision (Safayani et al., 2025).

Traditionally, CycleGAN (Zhu et al., 2017a) and its variants form the foundation for unpaired I2I translation (Choi et al., 2018; Park et al., 2020a; Fu et al., 2019; Zheng et al., 2022). These methods use bidirectional generators for domain translation, along with domain-specific discriminators. Training combines adversarial loss, guided by discriminators, and cycle consistency loss from the bidirectional structure. While CycleGAN-based methods have advanced unpaired I2I translation, they still rely on adversarial loss, which can cause instability, convergence issues, and mode collapse (Saad et al., 2024). Moreover, their performance lags behind that of modern generative models.

The recent emergence of diffusion models (DMs) has significantly advanced unpaired I2I translation, thanks to their exceptional generative capabilities through iterative denoising. SDEdit (Meng et al., 2022) performs image translation by solving the reverse SDE with a diffusion model trained on the target domain. This is achieved by introducing noise to the source image or mapping it to a noisy space using an inversion method (Wu & De la Torre, 2023). Additional regularizers balance the inherent realism-faithfulness tradeoff (Zhao et al., 2022; Sun et al., 2023).

On the other hand, Schrödinger Bridge (Schrödinger, 1932) offers a promising approach for translating between two arbitrary distributions using entropy-regularized optimal transport. Various methods have been developed for translating between data distributions, such as those proposed in (Wang et al., 2021; Chen et al., 2021; Liu et al., 2022; He et al., 2024), though many of these methods are

| Model | Single-step | Unpaired | Bi-direction | Discr. |
|---|---|---|---|---|
| SDEdit | ✗ | ✓ | ✗ | ✗ |
| EGSDE | ✗ | ✓ | ✗ | ✗ |
| CycleDiff | ✗ | ✓ | ✓ | ✗ |
| DDIB | ✗ | ✓ | ✓ | ✗ |
| DDBM | ✗ | ✗ | ✓ | ✗ |
| UNSB | ✗ | ✓ | ✗ | ✓ |
| CDBM | ✓ | ✗ | ✗ | ✗ |
| **IBCD (Ours)** | ✓ | ✓ | ✓ | ✗ |

Table 1: A systematic comparison of IBCD with other diffusion-based image-to-image translation models highlights several key advantages.

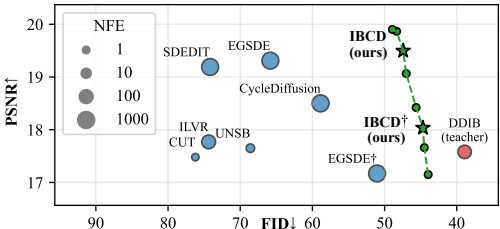

Figure 1: PSNR-FID trade-off comparison with baselines on the Cat→Dog task. Marker size represents NFE.

limited to paired settings. In contrast, DDIB (Su et al., 2023) addresses I2I translation by concatenating the ODE trajectories of two distinct DMs, making it suitable for unpaired settings, yet it still relies on numerous iterative steps. More recently, UNSB (Kim et al., 2024a) has been introduced to directly tackle unpaired I2I translation by regularizing Sinkhorn paths. Despite the aforementioned advancements in diffusion-based approaches, there still exist challenges encountered by the inference cost associated with their fundamental iterative nature, which limits their practical usability.

To address the limitations, we aim at the development of a *bidirectional single-step* generator that enables translation between two arbitrary distributions in *unpaired* settings without relying on adversarial losses (Tab. 1). Specifically, we propose Implicit Bridge Consistency Distillation (IBCD), a reformulation of the concept of consistency distillation (CD) (Song et al., 2023) that incorporates a diffusion implicit bridge model for translating between arbitrary data distributions. Unlike CD, which learns paths from Gaussian noise to data, IBCD connects trajectories from one arbitrary distribution to another one using a Probability Flow Ordinary Differential Equation (PF-ODE), allowing for flexible and efficient distribution translation.

However, simply extending CD can lead to increased distillation error due to error accumulation, as well as challenges related to model capacity and training scheme, which arise from integrating two trajectories and introducing bidirectionality. To address this, we propose a regularization method called Distribution Matching for Consistency Distillation (DMCD). Furthermore, we introduce a novel weighting scheme based on distillation difficulty, which applies a stronger DMCD penalty specifically to samples where the consistency loss alone proves insufficient. By integrating additional cycle translation loss with these advanced components, our approach significantly enhances the realism-faithfulness trade-off, achieving state-of-the-art performance in a single step, as shown in Fig. 1. The main contributions of our work are as follows:

- Propose Implicit Bridge Consistency Distillation (IBCD), a novel unpaired image translation framework that enables bidirectional translation with a single NFE and achieves state-of-the-art results.

- Introduce regularizing components, including Distribution Matching for Consistency Distillation (DMCD), an adaptive weighting scheme based on distillation difficulty, and cycle translation loss to mitigate inherent distillation errors.

- Demonstrate the effectiveness of IBCD through extensive experiments on toy, natural, and medical images, covering a range of diverse modalities.

## 2 PRELIMINARIES

### 2.1 IMAGE TRANSLATION WITH DIFFUSION MODELS

**Diffusion Models (DM).** In DMs (Ho et al., 2020; Song et al., 2021), the predefined forward process with the time variable $t \in [0, T]$ progressively corrupts data into pure Gaussian noise over a series of steps $T$. Specifically, given a data distribution $\mathbf{x}_0 \sim p(\mathbf{x}_0) := p_{\text{real}}(\mathbf{x})$, the distribution $\mathbf{x}_T \sim p(\mathbf{x}_T)$ approaches an isotropic normal distribution as noise is added according to the process $p(\mathbf{x}_t \mid \mathbf{x}_0) = \mathcal{N}(\mathbf{x}_0, t^2 \mathbf{I})$. The reverse of this process can be described by an SDE or a PF-ODE (Song et al., 2021) as follows:

$$\frac{d\mathbf{x}_t}{dt} = -t \nabla_{\mathbf{x}_t} \log p(\mathbf{x}_t) = \frac{\mathbf{x}_t - \mathbb{E}[\mathbf{x}_0|\mathbf{x}_t]}{t}, \tag{1}$$

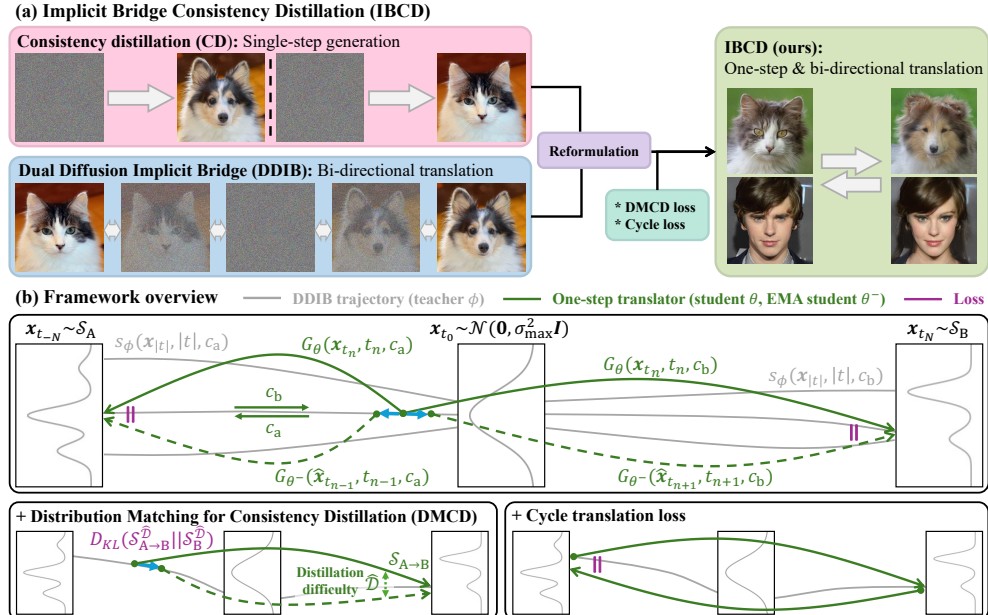

Figure 2: (a) IBCD performs single-step bi-directional translation using a distillation framework that reformulates consistency distillation with a diffusion implicit bridge and introduces regularizers. (b) The IBCD framework bridges two distributions by connecting the PF-ODE paths of two pre-trained diffusion models through bidirectionally extended consistency distillation. To mitigate distillation errors, we introduce distribution matching for consistency distillation and a cycle translation loss.

where the second equality follows from Tweedie's formula, $\mathbb{E}[\mathbf{x}_0|\mathbf{x}_t] = \mathbf{x}_t + t^2 \nabla_{\mathbf{x}_t} \log p(\mathbf{x}_t)$ (Efron, 2011; Kim & Ye, 2021). In practice, the neural network is trained to approximate the ground truth score function $\mathbf{s}_\phi(\mathbf{x}_t, t) \approx \nabla_{\mathbf{x}_t} \log p(\mathbf{x}_t)$ or the denoiser $D_\phi(\mathbf{x}_t, t) \approx \mathbb{E}[\mathbf{x}_0|\mathbf{x}_t]$ by denoising score matching (Vincent, 2011). By substituting the trained neural networks into Eq. (1), we can obtain the denoised sample by numerically integrating from $T$ to 0:

$$\mathbf{x}_0 = \mathbf{x}_T + \int_T^0 -t \cdot \mathbf{s}_\phi(\mathbf{x}, t) \, dt = \mathbf{x}_T + \int_T^0 \frac{\mathbf{x}_t - D_\phi(\mathbf{x}_t, t)}{t} \, dt. \tag{2}$$

To solve Eq. (2), an ODE solver, denoted as $\texttt{Solve}(\mathbf{x}_T; \phi, T, 0)$ (with an initial state $\mathbf{x}_T$ at time $T$ and ending at time 0, DM parameterized by $\phi$) can be applied. Examples include the Euler solver (Song et al., 2021; Ho et al., 2020), DPM-solver (Lu et al., 2022), or the second-order Heun solver (Karras et al., 2022). The sampling process typically requires dozens to hundreds of neural function evaluations (NFE) to effectively minimize discretization error during ODE solving.

**Dual Diffusion Implicit Bridge (DDIB).** DDIB (Su et al., 2023) is a simple yet effective method for I2I translation that leverages the connection between DMs and Schrödinger bridge problem (SBPs), where DMs act as implicit optimal transport models. DDIB requires training two individual DMs for the two domains A and B, denoted as $\mathbf{s}_{\phi^a}$ and $\mathbf{s}_{\phi^b}$. The sampling process involves sequential ODE solving as follows:

$$\mathbf{x}^l = \texttt{Solve}(\mathbf{x}^a; \phi^a, 0, T), \quad \mathbf{x}^b = \texttt{Solve}(\mathbf{x}^l; \phi^b, T, 0). \tag{3}$$

Here, $\mathbf{x}^l$ represents the latent code in the pure Gaussian noise domain, $\mathbf{x}^a$ is the image in the source domain, and $\mathbf{x}^b$ is the estimated image in the target domain. Thanks to the intermediate Gaussian distribution, DDIB automatically satisfies the cycle consistency property without any explicit regularization term (Zhu et al., 2017b; Choi et al., 2018).

## 2.2 EXISTING SINGLE-STEP ACCELERATION APPROACHES

**Consistency Distillation (CD).** The aim of the consistency distillation (CD) (Song et al., 2023) is to learn the direct mapping from noise to clean data. Specifically, the model is designed to predict $f_\theta(\mathbf{x}_t, t) = \mathbf{x}_0$, and is constrained to be *self-consistent*, meaning that outputs should be the same

for any time point input within the same PF-ODE trajectory, *i.e.*, $f(\mathbf{x}_t, t) = f(\mathbf{x}_{t'}, t')$ for all $t, t' \in [\epsilon, T]$, with the boundary condition $f_\theta(\mathbf{x}_\epsilon, \epsilon) = \mathbf{x}_\epsilon$. Here, $\epsilon$ is a small positive number, to avoid numerical instability at an $t = 0$. By discretizing the time interval $[\epsilon, T]$ into $N - 1$ sub-interval with boundaries $t_1 = \epsilon < t_2 < \cdots < t_N = T$, the resulting objective function for CD is given by:

$$\mathcal{L}_{\text{CD}}(\theta; \phi) = \mathbb{E}[\lambda(t_n) d(f_\theta(\mathbf{x}_{t_{n+1}}, t_{n+1}), f_{\theta^-}(\hat{\mathbf{x}}_{t_n}, t_n))], \tag{4}$$

where $n \sim \mathcal{U}[1, N - 1]$ and $\lambda(t_n)$ is weight hyperparameter, $d(\cdot, \cdot)$ measures the distance between two samples. $\theta^-$ is the exponential moving average (EMA) of the student parameter $\theta$, and $\phi$ represents the pre-trained teacher model, and $\mathcal{U}[\cdot]$ refers to the uniform distribution. The target $\hat{\mathbf{x}}_{t_n}$ is obtained by solving one-step ODE solver, *i.e.*, $\hat{\mathbf{x}}_{t_n} = \texttt{Solve}(\mathbf{x}_{t_{n+1}}; \phi, t_{n+1}, t_n)$, from $\mathbf{x}_{t_{n+1}} \sim \mathcal{N}(\mathbf{x}_0, t_{n+1}^2 \mathbf{I})$.

**Distribution Matching Distillation (DMD).** DMD (Yin et al., 2024; Wang et al., 2023) minimizes the Kullback-Leibler (KL) divergence between the real data distribution, $p^{\text{real}}$, and the student sample distribution, $p_\theta^{\text{fake}}$ to distill the diffusion model $\mathbf{s}_\phi^{\text{real}}$ into a single-step generator $f_\theta(\mathbf{x}_T) = \mathbf{x}_0$. Additionally, DMD introduces an auxiliary "fake" DM, $s_\psi^{\text{fake}}$, to approximate the score function of the student-generated sample distribution. This estimator is trained with denoising score matching, adapting in real-time as the student model progresses through training. The gradient of the DMD loss is then approximated as the difference between the two score functions:

$$\nabla_\theta D_{\text{KL}}(p_\theta^{\text{fake}} || p^{\text{real}}) \approx \nabla_\theta \mathcal{L}_{\text{DMD}} = \mathbb{E}_{\mathbf{x}_t, t, \mathbf{x}_T}[w_t(s_\psi^{\text{fake}}(\mathbf{x}_t, t) - s_\phi^{\text{real}}(\mathbf{x}_t, t))\nabla_\theta f_\theta(\mathbf{x}_T)], \tag{5}$$

where $\mathbf{x}_t \sim \mathcal{N}(f_\theta(\mathbf{x}_T), t^2 \mathbf{I})$, $t \sim \mathcal{U}(T_{\text{min}}, T_{\text{max}})$, $\mathbf{x}_T \sim \mathcal{N}(\mathbf{0}, T^2 \mathbf{I})$ and $w_t$ is a scalar weighting factor. DMD serves as an effective distillation loss that optimizes the student model from the view of the distribution, without relying on unstable adversarial loss (Goodfellow et al., 2014).

# 3 MAIN CONTRIBUTION

We aim to develop a one-step distillation method for bidirectional mapping between arbitrary distributions in an unpaired setting, using pretrained diffusion models. Specifically, given two domains $\mathcal{X}_{\text{A}}$ and $\mathcal{X}_{\text{B}}$ with unpaired datasets $\mathcal{S}_{\text{A}}$ and $\mathcal{S}_{\text{B}}$, our translator $f_\theta$ performs two translations: $f_\theta(\mathbf{x}^{\text{a}}, c_{\text{b}})$ for A→B and $f_\theta(\mathbf{x}^{\text{b}}, c_{\text{a}})$ for B→A, where $c_{\text{a}}$ and $c_{\text{b}}$ are class embeddings for the target domain. The concept of our method is illustrated in Fig. 2 by contrasting it to CD and DDIB. In the following, we describe a novel distillation approach with distribution matching, adaptive weighting, and a cycle loss for bidirectional reconstruction.

## 3.1 IMPLICIT BRIDGE CONSISTENT DISTILLATION

**Definition.** Our model architecture and diffusion process are based on the PF-ODE using EDM (Karras et al., 2022). To handle both domains with one generator, a pre-trained class conditional DMs, $\mathbf{s}_\phi(\mathbf{x}_t, t, c)$, is jointly trained for each domain with class conditions $c_{\text{a}}$ and $c_{\text{b}}$. Specifically, the teacher model $\mathbf{s}_\phi$ is trained using denoising score matching (DSM) for continuous-time $t = \sigma \sim \text{Lognormal} \in (0, \infty)$ without any modification from EDM. The timestep discretization for the sampling process is defined as $[t_0, t_1, \cdots, t_i, \cdots, t_N] = [\sigma_{\text{max}}, \sigma_{\text{max-1}}, \cdots, \sigma_{\text{min}}, 0]$. Since DDIB concatenates two independent ODEs into a single ODE, duplicated timesteps must be re-defined for consistency distillation (CD). We introduce a unique discretized timestep index $i$ and redefine the timestep $t$ for the concatenated trajectory ($\mathcal{X}_{\text{A}} \leftrightarrow \mathcal{X}_{\text{B}}$) as follows:

$$i = [\underbrace{-N, -N + 1, \cdots, -1}_{\mathcal{X}_{\text{A}}}, \underbrace{0}_{\mathcal{X}_{\text{A}} \cap \mathcal{X}_{\text{B}}}, \underbrace{1, \cdots, N - 1, N}_{\mathcal{X}_{\text{B}}}] \tag{6}$$

$$t_i = [-0, -\sigma_{\text{min}}, \cdots, -\sigma_{\text{max-1}}, +\sigma_{\text{max}}, +\sigma_{\text{max-1}}, \cdots, +\sigma_{\text{min}}, +0] \tag{7}$$

**Boundary Condition.** Given that the student model's output is enforced to be *self-consistent* with respect to timesteps in Eq. (7), we define the student as $f_\theta(\mathbf{x}_t, t, c)$, where $t$ is a non-zero real-valued timestep and $c \in c_a, c_b$ represents the target domain condition. For simplicity, we denote the opposite class embedding as $c'$, such that when $c = c_b$, $c' := c_a$. To enable bidirectional translation, we redefine the boundary condition to depend on the target domain condition $c$:

$$f(\mathbf{x}_{\epsilon(c)}, \epsilon(c), c) = \mathbf{x}_{\epsilon(c)}, \quad \text{where } \epsilon(c) = \begin{cases} t_{-N+1} &= -\sigma_{\text{min}}, & \text{for } c = c_a \\ t_{N-1} &= +\sigma_{\text{min}}, & \text{for } c = c_b \end{cases}. \tag{8}$$

This boundary condition, along with the IBCD loss introduced later, allows translation by injecting the desired domain condition: $f(\mathbf{x}_{\epsilon(c)}, \epsilon(c), c') = \mathbf{x}_{\epsilon(c')}$, where $f(\mathbf{x}_t, t, c_{\mathrm{b}})$ transforms $\mathbf{x}_t$ at any $t$ between $\mathcal{X}_{\mathrm{A}}$ and $\mathcal{X}_{\mathrm{B}}$ into a clean domain $\mathcal{X}_{\mathrm{B}}$ image $\mathbf{x}_{t_{N-1}}$ belonging to the same ODE trajectory, and vice versa. Since EDM/CD is not defined for negative $t$ values and does not directly align with our new boundary conditions, we introduce a non-trivial reformulation of the EDM/CD formulation, involving a novel parametrization tailored to the student model.[1] For more details on this extension, please refer to Appendix B (App. B).

**The Method.** To generate data pairs $(\mathbf{x}_{t_1}, \hat{\mathbf{x}}_{t_2})$ that lie on the same PF-ODE trajectory, we perform forward diffusion on the dataset and predict the next data point one step ahead using a suitable teacher model and ODE solver. For simplicity, we denote the teacher model $\phi$ conditioned on class $c$ as $\phi^c$. The data pair generation process in the direction of $\mathcal{X}_{\mathrm{A}} \to \mathcal{X}_{\mathrm{B}}$ (*i.e.* $c = c_{\mathrm{b}}$) for each domain is as follows:

$$\hat{\mathbf{x}}_{t_{i+1}} = \texttt{Solve}(\mathbf{x}_{t_i}; \phi^{\mathrm{a}}, |t_i|, |t_{i+1}|), \quad \hat{\mathbf{x}}_{t_j} = \texttt{Solve}(\mathbf{x}_{t_j}; \phi^{\mathrm{b}}, |t_j|, |t_{j+1}|), \qquad (9)$$

where $i \sim \mathcal{U}[-N+1, -1]$, $j \sim \mathcal{U}[0, N-2]$, $\mathbf{x}_{t_i} \sim \mathcal{N}(\mathbf{x}^{\mathrm{a}}, t_i^2 \mathbf{I})$, $\mathbf{x}_{t_j} \sim \mathcal{N}(\mathbf{x}^{\mathrm{b}}, t_j^2 \mathbf{I})$. Similarly, in the direction $\mathcal{X}_{\mathrm{B}} \to \mathcal{X}_{\mathrm{A}}$ (*i.e.* $c = c_{\mathrm{a}}$), the data pair for each domain can be generated as:

$$\hat{\mathbf{x}}_{t_{i-1}} = \texttt{Solve}(\mathbf{x}_{t_i}; \phi^{\mathrm{a}}, |t_i|, |t_{i-1}|), \quad \hat{\mathbf{x}}_{t_{j-1}} = \texttt{Solve}(\mathbf{x}_{t_j}; \phi^{\mathrm{b}}, |t_j|, |t_{j-1}|), \qquad (10)$$

where $i \sim \mathcal{U}[-N+2, 0]$, $j \sim \mathcal{U}[1, N-1]$. Given these distillation targets, our objective function of IBCD is defined as follows:

$$\mathcal{L}_{\mathrm{IBCD}}(\theta; \phi) = \mathop{\mathbb{E}}_{\mathbf{t}_1, \mathbf{x}_{\mathbf{t}_1}, c}[\lambda(\mathbf{t}_2) d(f_\theta(\mathbf{x}_{\mathbf{t}_1}, \mathbf{t}_1, c), f_{\theta^-}(\hat{\mathbf{x}}_{\mathbf{t}_2}, \mathbf{t}_2, c))], \qquad (11)$$

$$\text{where} \quad \mathbf{x}_{\mathbf{t}_1} = [\mathbf{x}_{t_i}; \mathbf{x}_{t_j}], \hat{\mathbf{x}}_{\mathbf{t}_2} = [\hat{\mathbf{x}}_{t_{i\pm1}}; \ \hat{\mathbf{x}}_{t_{j\pm1}}], \ c \in \mathcal{U}[\{c_{\mathrm{a}}, c_{\mathrm{b}}\}],$$

$$\mathbf{t}_1 = [t_i; t_j], \ \mathbf{t}_2 = [t_{i\pm1}; t_{j\pm1}], \ \theta^- = \texttt{sg}(\mu\theta^- + (1-\mu)\theta).$$

$n_{(\cdot)\pm1}$ denotes time index for each distillation direction in Eqs. (9), (10) and $\texttt{sg}$ indicates the stop-gradient operator. For a detailed explanation, see Algo. 1 in App. A.

Using a single domain-independent teacher model instead of two reduces memory and provides an effective initializer for the student model, acting as a unified model for both domains. By sharing the class condition in the teacher and the target domain condition in the student as a unified embedding, we can leverage the student's initialization weights, as $f(\mathbf{x}_t, t, c)$ is designed to output a clean image for domain $c$. This approach differs from methods in the literature (Kim et al., 2024b; Li & He, 2024), which extend CD in both directions or specify a target timestep, but don't fully integrate domain conditions into a cohesive framework.

## 3.2 OBJECTIVE FUNCTION

While our proposed vanilla IBCD lays a foundation for one-step bidirectional transport, there remain areas for refinement. First, the consistency loss is a local strategy (categorized by Kim et al. (2024b)), aligning consistency only with adjacent timesteps using the student's recursive output. This can accumulate local errors, leading to growing discrepancies between the student's prediction $f_\theta(\mathbf{x}_t, t, c)$ and the true boundary $\mathbf{x}_{\epsilon(c)}$, especially given IBCD's doubled trajectory.

Second, the student must handle bidirectional tasks and learn two distinct ODE trajectories. Although the teacher ODEs share timesteps, their differing output targets by target domain direction increase the complexity, placing additional demands on model capacity and making training more difficult, as also observed in CD by Li & He (2024). Finally, unlike EGSDE (Zhao et al., 2022), which balances realism and fidelity via expert weighting, vanilla IBCD lacks such a control mechanism, limiting its flexibility.

**Distribution Matching for Consistency Distillation.** To address these issues, we propose Distribution Matching for Consistency Distillation (DMCD), which extends the DMD loss to fit within the CD framework. DMCD builds on the DMD loss by optimizing the KL divergence between the student's output samples and the target domain data distributions across all timesteps in bidirectional tasks. Furthermore, it incorporates the distillation difficulty adaptive weighting factor $\hat{\mathcal{D}}(\cdot, \cdot)$. This

---

[1]Note this formulation applies exclusively to the student model.

adaptive weighting scheme helps to focus the optimization on challenging samples, thereby enhancing the overall performance and stability of the student model during training. The resulting DMCD is given by:

$$\nabla_\theta \mathcal{L}_{\text{DMCD}}(\theta; \phi, \psi) = \mathbb{E}_{\mathbf{t}_1, \mathbf{x}_{\mathbf{t}_1}, c, i, \mathbf{x}_{t_i}} \left[ w_{t_i} \hat{\mathcal{D}}(\text{sg}(\mathbf{x}_{\mathbf{t}_1}), c) \left( s_\psi(\mathbf{x}_{t_i}, t_i, c) - s_\phi(\mathbf{x}_{t_i}, t_i, c) \right) \nabla_\theta f_\theta(\mathbf{x}_{\mathbf{t}_1}, \mathbf{t}_1, c) \right]$$

$$\text{where} \quad i \sim \mathcal{U}[0, N-1], \ \mathbf{x}_{t_i} \sim \mathcal{N}(f_\theta(\mathbf{x}_{\mathbf{t}_1}, \mathbf{t}_1, c), t_i^2 \mathbf{I}) \tag{12}$$

where $\mathbf{t}_1, \mathbf{x}_{\mathbf{t}_1}, c$ are defined per from Eq. (11), and $w$ represents a time-dependent weighting factor introduced in DMD. The term $s_\psi(\mathbf{x}_t, t, c)$ denotes a class-conditional fake diffusion model, jointly trained via DSM on outputs of student $f_\theta$, adapting during training.

Unlike DMD, DMCD functions as a regularizer rather than the primary objective. This distinction is crucial in unpaired settings, where relying solely on the DMCD loss does not ensure a proper connection between two domains. Recently, a line of work (Rakitin et al., 2024) has similarly introduced DMD for I2I translation, where an additional $L_2$ regularization between the source and target is used to enforce domain mapping, while DMD itself focuses only on the target distribution matching. In contrast, IBCD constructs a trajectory between two distributions using consistency loss that enforces both domain mapping and target-distribution alignment, whereas DMCD handles the additional distribution matching component, serving as a regularizer to enhance realism. This integration allows for improved performance and stability without the drawbacks associated with adversarial training like Zhu et al. (2017b); Parmar et al. (2024); Kim et al. (2024a).

**Distillation Difficulty Adaptive Weighting.** DMCD effectively brings the translated distribution closer to the target data distribution, enhancing the realism of generated samples. However, this can also cause divergence from the teacher model's estimations, reducing faithfulness to the source domain. Ideally, DMCD should be applied more intensively to challenging PF-ODE trajectories that the student struggles to translate accurately, especially those involving source data near the decision boundary (App. D.1).

To address this, we propose a *distillation difficulty adaptive weighting* strategy. We define the concept of *distillation difficulty*, $\mathcal{D}([\mathbf{x}_{t_{-N+1}}, \cdots, \mathbf{x}_{t_{N-1}}], c) := d(f_\theta(\mathbf{x}_{\epsilon(c')}, \epsilon(c'), c), \mathbf{x}_{\epsilon(c)})$, which quantifies the challenge of distilling a given ODE trajectory generated by the teacher between domains. This allows DMCD to focus more aggressively on difficult trajectories, improving translation performance by targeting areas where the student struggles most. Such a strategy helps balance source faithfulness and realism by applying DMCD loss only where the IBCD loss is insufficient. However, estimating $\mathbf{x}_{\epsilon(c)}$ and $\mathbf{x}_{\epsilon(c')}$ from a given $\mathbf{x}_t$ requires at least $N$ NFEs with the teacher model for each DMCD loss calculation, which is computationally impractical. To address this, we propose a one-step approximation of the weighting factor $\mathcal{D}(\cdot, \cdot)$, defined:

$$\hat{\mathcal{D}}(\mathbf{x}_{\mathbf{t}_1}, c) = g(d(f_\theta(\mathbf{x}_{\mathbf{t}_1}, \mathbf{t}_1, c), \ f_{\theta^-}(\hat{\mathbf{x}}_{\mathbf{t}_2}, \mathbf{t}_2, c)))) \tag{13}$$

where $\mathbf{t}_1, \mathbf{t}_2, \mathbf{x}_{\mathbf{t}_1}, \hat{\mathbf{x}}_{\mathbf{t}_2}$ are defined in Eqs. (11), (12), and $g$ is any monotone increasing function. The validity of this weighting factor will be confirmed through experiments.

**Cycle Consistency Loss.** Similar to DDIB, our framework is designed to perform cycle translation and must therefore satisfy cycle consistency. The objective function of enforcing this requirement can be expressed as:

$$\mathcal{L}_{\text{cycle}}(\theta) = \mathbb{E}_{c, \mathbf{x}_{\epsilon(c)}} \left[ d(f_\theta(f_\theta(\mathbf{x}_{\epsilon(c)}, \epsilon(c), c'), \epsilon(c'), c), \mathbf{x}_{\epsilon(c)}) \right]. \tag{14}$$

**Final Loss Functions.** The final loss, weighted by $\lambda_{\text{DMCD}}, \lambda_{\text{cycle}}$, for training $f_\theta$ is given by:

$$\theta^* = \arg\min_\theta \mathcal{L}_{\text{IBCD}} + \lambda_{\text{DMCD}} \mathcal{L}_{\text{DMCD}} + \lambda_{\text{cycle}} \mathcal{L}_{\text{cycle}}. \tag{15}$$

Empirically, we found that the following adaptive training strategy further improves performance: the training process begins with only the IBCD loss; as the student model approaches convergence, the DMCD and cycle consistency losses are additionally introduced to further refine the model's performance. The detailed final algorithm can be found in Algo. 2, App. A.

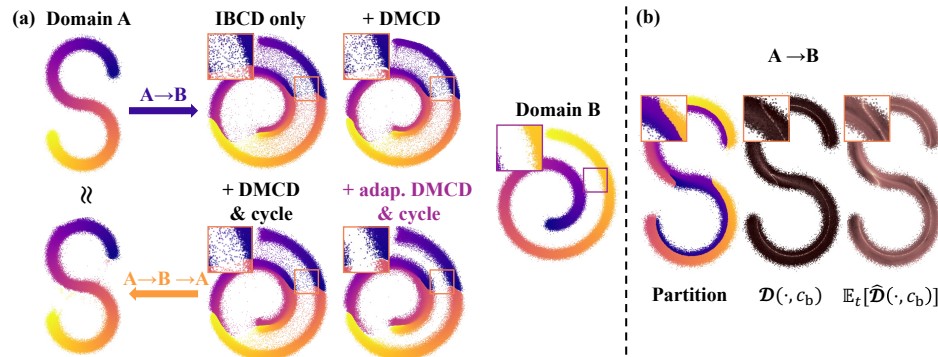

Figure 3: (a) Bidirectional translation results on a toy dataset, showing the contributions of each component. (b) Visualization of distillation difficulty $\mathcal{D}(\cdot, c_b)$ and its one-step approximation $\mathbb{E}_t[\hat{\mathcal{D}}(\cdot, c_b)]$ for A→B translation, with $g$ selected as a logarithm.

## 4 EXPERIMENTS

### 4.1 TOY DATA EXPERIMENT

To evaluate the effectiveness of our framework in a controlled setting, we conducted bidirectional translation experiments on a two-dimensional synthetic toy dataset, where the domains $A$ and $B$ were represented by the S-curve and Swiss roll distributions, respectively.

**Validity of the IBCD.** Fig. 3(a) shows the translation results from domain A→B for various models, highlighting the cumulative effectiveness of each component in our framework. Distillation with only the IBCD loss achieves basic translation but incorrectly maps some points to low-density regions of the target domain, particularly from the source domain's decision boundaries (Appendix D.1). Adding the DMCD loss improves translation by guiding more points toward high-density regions, but it fails to reposition points in low-density areas and reduces mode coverage by pushing points in high-density regions even further. Introducing a cycle loss alleviates the reduction in mode coverage caused by DMCD and refines the decision boundaries in the target domain. Finally, incorporating distillation difficulty adaptive weighting into DMCD selectively corrects points that have drifted into low-density regions, guiding them toward higher-density areas. The complete cycle translation (A→B→A) using a model trained with our final approach effectively demonstrates cycle consistency, validating the robustness and fidelity of our method.

**Distillation Difficulty.** Fig. 3(b) illustrates the impact of distillation difficulty on the translation process. On the left, we show the decision boundary of the source domain resulting from the translation from the target to the source domain by the DDIB teacher model. The middle and right panels depict $\mathcal{D}([\mathbf{x}_{t_{-N+1}}, \cdots, \mathbf{x}_{t_{N-1}}], c_b)$ and its expected one-step approximation, $\mathbb{E}_{t \sim \mathcal{U}[-N+1, N-2]}[\hat{\mathcal{D}}(\mathbf{x}_t, c_b)]$ for the A→B translation, plotted at the source domain location $\mathbf{x}_{\epsilon(c_a)}$. The distillation difficulty measure effectively captures the decision boundary, indicating challenging regions for the student model. As shown, it's one-step approximation provides an accurate and suitable representation of the distillation difficulty, demonstrating its utility in guiding the training process and improving translation accuracy.

### 4.2 UNPAIRED IMAGE-TO-IMAGE TRANSLATION

In this section, we apply IBCD to various I2I translation tasks, our primary focus. We evaluate its performance across these tasks to demonstrate effectiveness and robustness.

**Evaluation.** Following EGSDE's evaluation protocol, a widely used benchmark for unpaired I2I tasks, we tested our method on Cat→Dog, Wild→Dog (AFHQ) (Choi et al., 2020), and Male→Female (CelebA-HQ) (Karras, 2018). We trained AFHQ and CelebA-HQ DMs as teacher models. Single-step translation models for Cat↔Dog and Wild↔Dog were distilled from the AFHQ DM, and Male↔Female from the CelebA-HQ DM. We used FID (Heusel et al., 2017) and Density-Coverage (Naeem et al., 2020) for translation reality, and PSNR, SSIM (Wang et al., 2004),

LPIPS (Zhang et al., 2018), and CLIP score (Hessel et al., 2021) for translation faithfulness. A user study was also conducted to evaluate perceptual quality and human preference.

**Comparison results.** Fig. 1, Fig. 4, and Tab. 2 show that IBCD consistently outperforms baseline models in both qualitative and quantitative comparisons. IBCD strikes a balance between faithfulness and realism, while IBCD[†] emphasizes realism. These results demonstrate our effectiveness in improving the faithfulness-realism trade-off across tasks and metrics. User studies and perceptual metrics (App. D.2) further confirm the superiority of our method in terms of human preference. Although the student model shows reduced realism compared to the teacher, it exhibits improved faithfulness. The decline in realism may stem from distillation errors, which could be caused by previously discussed factors such as the single-step conversion process. Unlike the teacher, the student model integrates information from both domains, possibly leading it to prioritize faithfulness. Interestingly, in some cases, the student's samples surpass the teacher's in realism, likely due to auxiliary losses beyond the IBCD loss. This suggests that the student's ability to combine domain information and auxiliary training components can enhance overall performance.

**Ablation Study.** We conducted an ablation study on the Cat→Dog task to evaluate the effectiveness of each component. In this study, DMCD loss, cycle translation loss, and distillation difficulty adap-

Table 2: **Quantitative comparison of unpaired image-to-image translation tasks**. Most results are from the EGSDE paper, except those marked with *, which are from our re-implementation and Density-Coverage metric (Naeem et al., 2020). Marker † indicates a hyperparameter configuration that prioritizes realism over faithfulness.

| Method | NFE ↓ | FID ↓ | PSNR ↑ | SSIM ↑ | Density ↑ | Coverage ↑ |
|---|---|---|---|---|---|---|
| **Cat→Dog** | | | | | | |
| CycleGAN (Zhu et al., 2017b) | 1 | 85.9 | - | - | - | - |
| Self-Distance (Benaim & Wolf, 2017) | 1 | 144.4 | - | - | - | - |
| GcGAN (Fu et al., 2019) | 1 | 96.6 | - | - | - | - |
| LeSeSIM (Zheng et al., 2021) | 1 | 72.8 | - | - | - | - |
| StarGAN v2 (Choi et al., 2020) | 1 | $54.88 \pm 1.01$ | $10.63 \pm 0.10$ | $0.270 \pm 0.003$ | - | - |
| CUT (Park et al., 2020b) | 1 | 76.21 | 17.48 | 0.601 | 0.971 | 0.696 |
| UNSB* (Kim et al., 2024a) | 5 | 68.59 | 17.65 | 0.587 | 1.045 | 0.706 |
| ILVR (Choi et al., 2021) | 1000 | $74.37 \pm 1.55$ | $17.77 \pm 0.02$ | $0.363 \pm 0.001$ | $1.019 \pm 0.030$ | $0.566 \pm 0.012$ |
| SDEdit (Meng et al., 2022) | 1000 | $74.17 \pm 1.01$ | $19.19 \pm 0.01$ | $0.423 \pm 0.001$ | $0.997 \pm 0.021$ | $0.526 \pm 0.014$ |
| EGSDE (Zhao et al., 2022) | 1000 | $65.82 \pm 0.77$ | $19.31 \pm 0.02$ | $0.415 \pm 0.001$ | $1.258 \pm 0.027$ | $0.634 \pm 0.023$ |
| EGSDE† (Zhao et al., 2022) | 1200 | $51.04 \pm 0.37$ | $17.17 \pm 0.02$ | $0.361 \pm 0.001$ | $1.509 \pm 0.038$ | $0.823 \pm 0.021$ |
| CycleDiffusion (Wu & De la Torre, 2023) | 1000(+100) | $58.08 \pm 1.08$ | $18.36 \pm 0.04$ | $0.537 \pm 0.001$ | $0.905 \pm 0.023$ | $0.767 \pm 0.028$ |
| SDDM (Sun et al., 2023) | 100 | $62.29 \pm 0.63$ | - | $0.422 \pm 0.001$ | - | - |
| SDDM† (Sun et al., 2023) | 120 | $49.43 \pm 0.23$ | - | $0.361 \pm 0.001$ | - | - |
| GPT-Image-1 (Foundation) (OpenAI, 2025) | ≫1 | 77.81 | 12.18 | 0.283 | 0.947 | 0.586 |
| DDIB* (Teacher) (Su et al., 2023) | 160 | 38.91 | 17.58 | 0.588 | 1.528 | 0.934 |
| **IBCD (Ours)** | 1 | $47.44 \pm 0.03$ | $19.50 \pm 3e\text{-}4$ | $0.701 \pm 1e\text{-}5$ | $1.412 \pm 0.007$ | $0.940 \pm 0.003$ |
| **IBCD† (Ours)** | 1 | $44.77 \pm 0.07$ | $18.04 \pm 2e\text{-}4$ | $0.663 \pm 8e\text{-}6$ | $1.542 \pm 0.005$ | $0.935 \pm 0.003$ |
| **Wild→Dog** | | | | | | |
| CUT (Park et al., 2020b) | 1 | 92.94 | 17.20 | 0.592 | - | - |
| UNSB* (Kim et al., 2024a) | 5 | 70.03 | 16.86 | 0.573 | 1.035 | 0.704 |
| ILVR (Choi et al., 2021) | 1000 | $75.33 \pm 1.22$ | $16.85 \pm 0.02$ | $0.287 \pm 0.001$ | $1.275 \pm 0.046$ | $0.531 \pm 0.013$ |
| SDEdit (Meng et al., 2022) | 1000 | $68.51 \pm 0.65$ | $17.98 \pm 0.01$ | $0.343 \pm 0.001$ | $1.292 \pm 0.045$ | $0.636 \pm 0.018$ |
| EGSDE (Zhao et al., 2022) | 1000 | $59.75 \pm 0.62$ | $18.14 \pm 0.02$ | $0.343 \pm 0.001$ | $1.482 \pm 0.018$ | $0.683 \pm 0.013$ |
| EGSDE† (Zhao et al., 2022) | 1200 | $50.43 \pm 0.52$ | $16.40 \pm 0.01$ | $0.300 \pm 0.001$ | $1.733 \pm 0.022$ | $0.782 \pm 0.014$ |
| CycleDiffusion (Wu & De la Torre, 2023) | 1000(+100) | $58.92 \pm 0.72$ | $17.68 \pm 0.03$ | $0.458 \pm 0.001$ | $1.014 \pm 0.034$ | $0.801 \pm 0.027$ |
| SDDM (Sun et al., 2023) | 100 | $57.38 \pm 0.53$ | - | $0.328 \pm 0.001$ | - | - |
| GPT-Image-1 (Foundation) (OpenAI, 2025) | ≫1 | 100.72 | 12.38 | 0.230 | 0.578 | 0.294 |
| DDIB* (Teacher) (Su et al., 2023) | 160 | 38.59 | 17.03 | 0.552 | 1.594 | 0.924 |
| **IBCD (Ours)** | 1 | $48.60 \pm 0.11$ | $18.25 \pm 2e\text{-}4$ | $0.653 \pm 2e\text{-}5$ | $1.539 \pm 0.006$ | $0.921 \pm 0.005$ |
| **IBCD† (Ours)** | 1 | $46.06 \pm 0.06$ | $16.78 \pm 1e\text{-}4$ | $0.612 \pm 1e\text{-}5$ | $1.583 \pm 0.010$ | $0.919 \pm 0.004$ |
| **Male→Female** | | | | | | |
| CUT (Park et al., 2020b) | 1 | 31.94 | 19.87 | 0.74 | - | - |
| UNSB* (Kim et al., 2024a) | 5 | 28.62 | 19.55 | 0.687 | 0.576 | 0.635 |
| ILVR (Choi et al., 2021) | 1000 | $46.12 \pm 0.33$ | $18.59 \pm 0.02$ | $0.510 \pm 0.001$ | - | - |
| SDEdit (Meng et al., 2022) | 1000 | $49.43 \pm 0.47$ | $20.03 \pm 0.01$ | $0.572 \pm 0.000$ | $0.782 \pm 0.020$ | $0.380 \pm 0.018$ |
| EGSDE (Zhao et al., 2022) | 1000 | $41.93 \pm 0.11$ | $20.35 \pm 0.01$ | $0.574 \pm 0.000$ | $0.875 \pm 0.032$ | $0.437 \pm 0.017$ |
| EGSDE† (Zhao et al., 2022) | 1200 | $30.61 \pm 0.22$ | $18.32 \pm 0.02$ | $0.510 \pm 0.001$ | $0.955 \pm 0.019$ | $0.621 \pm 0.016$ |
| SDDM (Sun et al., 2023) | 100 | $44.37 \pm 0.23$ | - | $0.526 \pm 0.001$ | - | - |
| GPT-Image-1 (Foundation) (OpenAI, 2025) | ≫1 | 60.50 | 13.11 | 0.381 | 0.503 | 0.269 |
| DDIB* (Teacher) (Su et al., 2023) | 160 | 23.69 | 18.70 | 0.664 | 0.969 | 0.808 |
| **IBCD (Ours)** | 1 | $24.93 \pm 0.03$ | $20.51 \pm 4e\text{-}4$ | $0.749 \pm 3e\text{-}5$ | $1.160 \pm 0.008$ | $0.814 \pm 0.006$ |
| **IBCD† (Ours)** | 1 | $24.70 \pm 0.03$ | $20.11 \pm 4e\text{-}4$ | $0.744 \pm 3e\text{-}5$ | $1.145 \pm 0.003$ | $0.815 \pm 0.004$ |

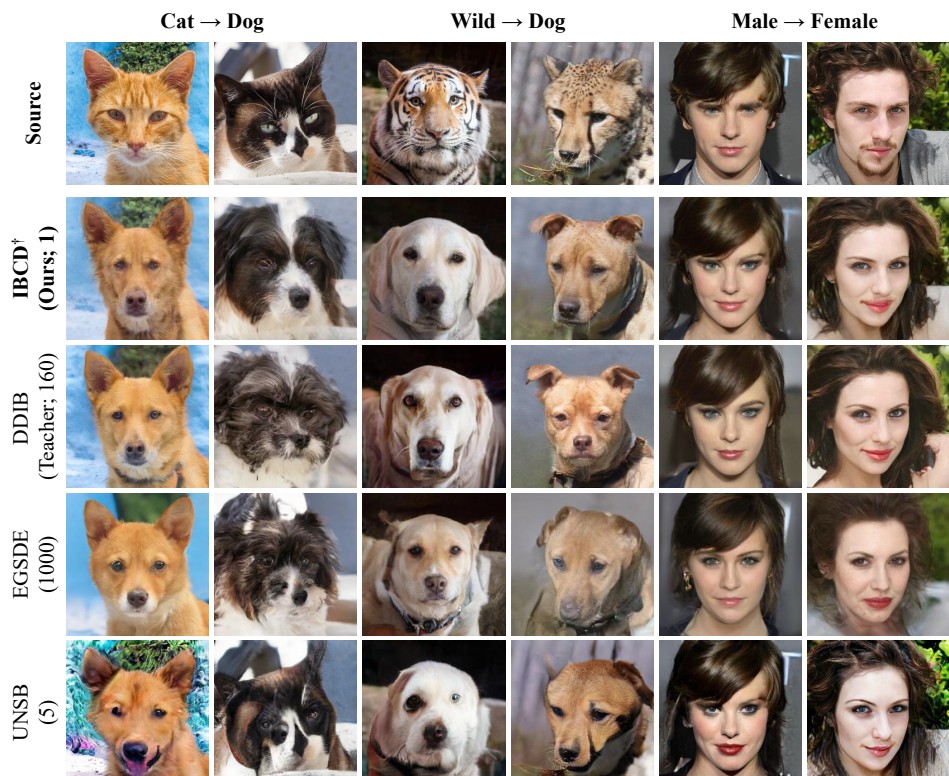

Figure 4: **Qualitative comparison of unpaired image-to-image translation tasks**. Compared to other baselines, our model achieves more realistic and source-faithful translations in a single step. The numbers in parentheses represent inference NFE.

Table 3: Quantitative ablation study results in the Cat→Dog task *under the lowest FID* (similar FID condition except for the vanilla IBCD).

| Component | FID↓ | PSNR ↑ |
|---|---|---|
| IBCD only | 48.12 | 18.27 |
| + DMCD | 44.40 | 17.95 |
| + DMCD & Cycle | 44.31 | 18.22 |
| **+ adaptive DMCD & Cycle** | **44.69** | **18.97** |

tive weighting (adaptive DMCD) were sequentially added to the baseline IBCD loss-only model. To assess distillation error, we calculated PSNR relative to the DDIB teacher. Tab. 3 and Fig. 7 display the results for each ablated model that *achieved the lowest FID*. In Tab. 3, each component significantly reduces FID beyond the lower bound achieved by vanilla IBCD, while minimizing PSNR degradation due to the task's inherent trade-off and reducing distillation error. Adaptive DMCD has been particularly effective when prioritizing the lowest FID in the trade-off curve, significantly reducing distillation errors as well. These findings confirm that the components of IBCD work synergistically to improve the balance between faithfulness and realism. In addition, the results in Fig. 7 similarly demonstrate that DMCD enhances the realism of the generated images, while the cycle loss and adaptive DMCD loss qualitatively improve source faithfulness (indicated by the white arrows).

For additional experimental content, including toy datasets, medical images, user studies, auxiliary loss analysis, and other details not covered in the main text, please refer to App. D.

## 5 CONCLUSION

In this work, we introduced IBCD, a novel unpaired bidirectional single-step image translation framework. By distilling the diffusion implicit bridge through a novelly parametrized reformulation CD framework, we achieved bidirectional translation without paired data or adversarial training.

Our approach overcomes traditional CD limitations with DMCD and distillation difficulty adaptive weighting strategies. Empirical evaluations on toy and high-dimensional datasets demonstrate IBCD's effectiveness and scalability. We believe IBCD represents a significant advancement in general single-step image translation, offering a versatile and efficient solution for various image tasks, particularly in scenarios with limited paired data and those where low latency is crucial.

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

SUPPLEMENTARY MATERIAL

# A  ALGORITHMS

In this section, we present the vanilla implicit bridge consistency distillation algorithm (Algo. 1), which utilizes only the IBCD losses. Additionally, we introduce the final implicit bridge consistency distillation algorithm (Algo. 2), which incorporates all the losses discussed in the text, including DMCD, adaptive weighting strategies, and cycle translation loss, to enhance performance and address the limitations identified in the vanilla version.

---

**Algorithm 1:** (Vanilla) Implicit Bridge Consistent Distillation (IBCD)

---

**Input:** Teacher diffusion model $\phi$, datasets $\mathcal{S}_A$ and $\mathcal{S}_B$, class conditions $c_a$ and $c_b$.

1  $j \leftarrow 0, \theta \leftarrow \phi, \theta^- \leftarrow \phi$

2  **repeat**

3      $c \leftarrow$ **if** $(j\%2 == 0$ **then** $c_a$ **else** $c_b)$

4      Sample $\mathbf{x}^a \sim \mathcal{S}_A$, $\mathbf{x}^b \sim \mathcal{S}_B$

5      **if** $c == c_b$ **then**

6          Sample $i \sim \mathcal{U}[-N+1, -1]$, $j \sim \mathcal{U}[0, N-2]$

7      **else**

8          Sample $i \sim \mathcal{U}[-N+2, 0]$, $j \sim \mathcal{U}[1, N-1]$

9      Sample $\mathbf{x}_{t_i} \sim \mathcal{N}(\mathbf{x}^a, t_i^2 \mathbf{I})$, $\mathbf{x}_{t_j} \sim \mathcal{N}(\mathbf{x}^b, t_j^2 \mathbf{I})$

10      **if** $c == c_b$ **then**

11          Estimate $\hat{\mathbf{x}}_{t_{i+1}}, \hat{\mathbf{x}}_{t_{j+1}}$ with Eq. (9)

12      **else**

13          Estimate $\hat{\mathbf{x}}_{t_{i-1}}, \hat{\mathbf{x}}_{t_{j-1}}$ with Eq. (10)

14      $\mathbf{t}_1 \leftarrow [t_i; t_j]$, $\mathbf{t}_2 = [t_{i\pm1}; t_{j\pm1}]$

15      $\mathbf{x}_{\mathbf{t}_1} \leftarrow [\mathbf{x}_{t_i}; \mathbf{x}_{t_j}]$, $\hat{\mathbf{x}}_{\mathbf{t}_2} \leftarrow [\hat{\mathbf{x}}_{t_{i\pm1}}; \hat{\mathbf{x}}_{t_{j\pm1}}]$

16      $\mathcal{L}_{\text{IBCD}} \leftarrow [\lambda(\mathbf{t}_2) d_{\text{IBCD}}(f_\theta(\mathbf{x}_{\mathbf{t}_1}, \mathbf{t}_1, c), f_{\theta^-}(\hat{\mathbf{x}}_{\mathbf{t}_2}, \mathbf{t}_2, c))]$

17      $\theta \leftarrow \theta - \zeta_\theta \nabla_\theta \mathcal{L}_{\text{IBCD}}$

18      $\theta^- \leftarrow \text{sg}(\mu\theta^- + (1-\mu)\theta)$

19      $j \leftarrow j + 1$

20  **until** $\mathcal{L}_{\text{IBCD}}$ *convergence*;

**Output:** Unified single-step model $f_\theta$ for bidirectional image translation.

---

---

**Algorithm 2:** (Final) Implicit Bridge Consistent Distillation (IBCD)

---

**Input:** Teacher diffusion model $\phi$, datasets $\mathcal{S}_A$ and $\mathcal{S}_B$, class conditions $c_a$ and $c_b$.

1   $j \leftarrow 0, \theta \leftarrow \phi, \theta^- \leftarrow \phi, \psi \leftarrow \phi$

2   **repeat**

3     $c \leftarrow$ **if** $(j\%2 == 0$ **then** $c_a$ **else** $c_b)$

4     Sample $\mathbf{x}^a \sim \mathcal{S}_A,\ \mathbf{x}^b \sim \mathcal{S}_B$

     //

     // IBCD loss

5     **if** $c == c_b$ **then**

6       $\lfloor$ Sample $i \sim \mathcal{U}[-N + 1, -1],\ j \sim \mathcal{U}[0, N - 2]$

7     **else**

8       $\lfloor$ Sample $i \sim \mathcal{U}[-N + 2, 0],\ j \sim \mathcal{U}[1, N - 1]$

9     Sample $\mathbf{x}_{t_i} \sim \mathcal{N}(\mathbf{x}^a, t_i^2\mathbf{I}),\ \mathbf{x}_{t_j} \sim \mathcal{N}(\mathbf{x}^b, t_j^2\mathbf{I})$

10    **if** $c == c_b$ **then**

11      $\lfloor$ Estimate $\hat{\mathbf{x}}_{t_{i+1}},\ \hat{\mathbf{x}}_{t_{j+1}}$ with Eq. (9)

12    **else**

13      $\lfloor$ Estimate $\hat{\mathbf{x}}_{t_{i-1}},\ \hat{\mathbf{x}}_{t_{j-1}}$ with Eq. (10)

14    $\mathbf{t}_1 \leftarrow [t_i; t_j],\ \mathbf{t}_2 = [t_{i\pm 1}; t_{j\pm 1}]$

15    $\mathbf{x}_{\mathbf{t}_1} \leftarrow [\mathbf{x}_{t_i}; \mathbf{x}_{t_j}],\ \hat{\mathbf{x}}_{\mathbf{t}_2} \leftarrow [\hat{\mathbf{x}}_{t_{i\pm 1}}; \hat{\mathbf{x}}_{t_{j\pm 1}}]$

16    $\mathcal{L}_{\text{IBCD}} \leftarrow [\lambda(\mathbf{t}_2)d_{\text{IBCD}}(f_\theta(\mathbf{x}_{\mathbf{t}_1}, \mathbf{t}_1, c), f_{\theta^-}(\hat{\mathbf{x}}_{\mathbf{t}_2}, \mathbf{t}_2, c))]$

     //

     // DMCD loss

17    Sample $i \sim \mathcal{U}[0, N - 1]$

18    Sample $\mathbf{x}_{t_i} \sim \mathcal{N}(f_\theta(\mathbf{x}_{\mathbf{t}_1}, \mathbf{t}_1, c), t_i^2\mathbf{I})$

19    $\hat{\mathcal{D}} \leftarrow \text{sg}(g(d_{\text{DMCD}}(f_\theta(\mathbf{x}_{\mathbf{t}_1}, \mathbf{t}_1, c), f_{\theta^-}(\hat{\mathbf{x}}_{\mathbf{t}_2}, \mathbf{t}_2, c))))$

20    $\nabla_\theta \mathcal{L}_{\text{DMCD}} \leftarrow w_{t_i}\hat{\mathcal{D}} \cdot (s_\psi(\mathbf{x}_{t_i}, t_i, c) - s_\phi(\mathbf{x}_{t_i}, t_i, c))\nabla_\theta f_\theta(\mathbf{x}_{\mathbf{t}_1}, \mathbf{t}_1, c)$

     //

     // Cycle loss

21    Sample $\mathbf{x}_{\epsilon(c_a)} \sim \mathcal{N}(\mathbf{x}^a, \sigma_{\min}^2\mathbf{I}),\ \mathbf{x}_{\epsilon(c_b)} \sim \mathcal{N}(\mathbf{x}^b, \sigma_{\min}^2\mathbf{I})$

22    $\mathbf{t}_3 \leftarrow [\epsilon(c_a); \epsilon(c_b)],\ \mathbf{t}_4 \leftarrow [\epsilon(c_b); \epsilon(c_a)]$

23    $\mathbf{c}_3 \leftarrow [c_b; c_a],\ \mathbf{c}_4 \leftarrow [c_a; c_b]$

24    $\mathbf{x}_{\mathbf{t}_3} \leftarrow [\mathbf{x}_{\epsilon(c_a)}; \mathbf{x}_{\epsilon(c_b)}]$

25    $\mathcal{L}_{\text{cycle}} \leftarrow d_{\text{cycle}}(f_\theta(f_\theta(\mathbf{x}_{\mathbf{t}_3}, \mathbf{t}_3, \mathbf{c}_3), \mathbf{t}_4, \mathbf{c}_4), \mathbf{x}_{\mathbf{t}_3})$

     //

     // Optimize the student

26    $\nabla_\theta \mathcal{L}_{\text{total}} \leftarrow \nabla_\theta \mathcal{L}_{\text{IBCD}} + \lambda_{\text{DMCD}}\nabla_\theta \mathcal{L}_{\text{DMCD}} + \lambda_{\text{cycle}}\nabla_\theta \mathcal{L}_{\text{cycle}}$

27    $\theta \leftarrow \theta - \zeta_\theta \nabla_\theta \mathcal{L}_{\text{total}}$

28    $\theta^- \leftarrow \text{sg}(\mu\theta^- + (1 - \mu)\theta)$

     //

     // Optimize the fake DM

29    $\mathcal{L}_{DSM} \leftarrow$ DSM loss of EDM with sample $f_\theta(\mathbf{x}_{\mathbf{t}_1}, \mathbf{t}_1, c)$, class condition $c$, and fake DM $\phi$

30    $\phi \leftarrow \phi - \zeta_\phi \nabla_\phi \mathcal{L}_{\text{DSM}}$

31    $j \leftarrow j + 1$

32   **until** $\mathcal{L}_{total}$ *convergence*;

**Output:** Unified single-step model $f_\theta$ for bidirectional image translation.

---

## B EXTENDING EDM/CD FOR THE IBCD

**Model Parametrization.** The EDM (Karras et al., 2022) parametrization for the student $f_\theta$ in consistency distillation (Song et al., 2023) is defined as follows for positive real-valued $t$ and the neural network $F_\theta$:

$$f_\theta(\mathbf{x}_t, t) = c_{\text{skip}}(t)\mathbf{x}_t + c_{\text{out}}(t)F_\theta(c_{\text{in}}(t)\mathbf{x}_t, t'(t)). \quad (16)$$

In CD, authors choose

$$c_{\text{skip}}(t) = \frac{\sigma_{\text{data}}^2}{(t-\epsilon)^2 + \sigma_{\text{data}}^2}, \qquad c_{\text{out}}(t) = \frac{\sigma_{\text{data}}(t-\epsilon)}{\sqrt{\sigma_{\text{data}}^2 + t^2}}, \qquad c_{\text{in}}(t) = \frac{1}{\sqrt{\sigma_{\text{data}}^2 + t^2}}, \quad (17)$$

$$t'(t) = 250 \cdot \ln(t + 10^{-44})$$

to satisfies the boundary condition $f(\mathbf{x}_\epsilon, \epsilon) = \mathbf{x}_\epsilon$, and rescales the timestep.

For IBCD, we parametrize the student $f_\theta$ for non-zero real-valued $t$ and target domain condition $c$ as:

$$f_\theta(\mathbf{x}_t, t, c) = c_{\text{skip}}(t, c)\mathbf{x}_t + c_{\text{out}}(t, c)F_\theta(c_{\text{in}}(t, c)\mathbf{x}_t, t'(t)), \quad (18)$$

which reflects the necessity for $c_{\text{skip}}, c_{\text{out}}$, and $c_{\text{in}}$ depend on target domain condition $c$, ensuring that the proper boundary conditions can be applied at $t = \epsilon(c)$ depending on the target domain $c \in \{c_a, c_b\}$ direction.

Although the student model is fully trained during the distillation process and does not theoretically need to be compatible with the teacher model, initializing it using the teacher model makes it advantageous to design the student to be as compatible as possible. We select $c_{\text{skip}}, c_{\text{out}}$, and $c_{\text{in}}$ according to Eq. (19), (20), (21), ensuring continuity and compliance with the new boundary conditions while maintaining the definitions within the target domain regions ($t > 0$ for $c = c_b$, $t < 0$ for $c = c_a$).

$$c_{\text{skip}}(t, c) = \begin{cases} \frac{1+\text{sign}(t)}{2}\frac{\sigma_{\text{data}}^2}{(t-\epsilon(c))^2+\sigma_{\text{data}}^2} & \text{if } c = c_b \\ \frac{1+\text{sign}(-t)}{2}\frac{\sigma_{\text{data}}^2}{(t-\epsilon(c))^2+\sigma_{\text{data}}^2} & \text{if } c = c_a \end{cases} \quad (19)$$

$$c_{\text{out}}(t, c) = \begin{cases} \frac{1+\text{sign}(t)}{2}\frac{\sigma_{\text{data}}(t-\epsilon(c))}{\sqrt{\sigma_{\text{data}}^2+t^2}} + \frac{1-\text{sign}(t)}{2}\sigma_{\text{data}} & \text{if } c = c_b \\ -\frac{1+\text{sign}(-t)}{2}\frac{\sigma_{\text{data}}(t-\epsilon(c))}{\sqrt{\sigma_{\text{data}}^2+t^2}} + \frac{1-\text{sign}(-t)}{2}\sigma_{\text{data}} & \text{if } c = c_a \end{cases} \quad (20)$$

$$c_{\text{in}}(t, c) = \frac{1}{\sqrt{\sigma_{\text{data}}^2 + t^2}} \quad (21)$$

We also extend the timestep rescaler as Eq. (22) to a symmetric and continuous form, ensuring shape compatibility with the original positive-bound domain. This symmetric design reflects the fact that the sign of the timestep separates the domains, while its absolute value represents the noise magnitude:

$$t'(t) = 250 \cdot \text{sign}(t)(\ln(|t| + 10^{-3}) - \ln(\sigma_{\text{max}} + 10^{-44})). \quad (22)$$

This approach preserves the structural integrity of the model and maintains consistent behavior across both domains. The parametrization extension of EDM/CD, as presented here, is visually illustrated in Fig. 5.

**Non-differentiability of the ODE Path.** In this paragraph, we address the behavior of IBCD at the continuous but non-differentiable point at the center of the PF-ODE trajectory ($i = 0$). Despite the introduction of this point, we show that Theorem 1 from Song et al. (2023) (its App. A.2) remains applicable, thereby proving the validity of the consistency distillation framework. First, the Lipschitz condition for $f_\theta(x_t, t)$ continues to hold. The primary distinction between IBCD and CD occurs in the $t$ direction, and so we focus on this aspect. The output of $f_\theta$, which predicts the clean target domain image, remains constant along a given PF-ODE trajectory, independent of $t$. As a result, the Lipschitz condition is not impacted by the non-differentiable point, as the trajectory is continuous. Since the change in the $x_t$ direction is equivalent to that in the CD framework, the Lipschitz continuity assumption from CD is still valid.

Next, we consider the local truncation error of the ODE solver. The non-differentiable point is appropriately captured by our discretization scheme. At this point, the gradient is a combination of gradients from both sides of the trajectory, ensuring stable numerical integration. For example, using an Euler solver in the forward direction (from domain A to domain B):

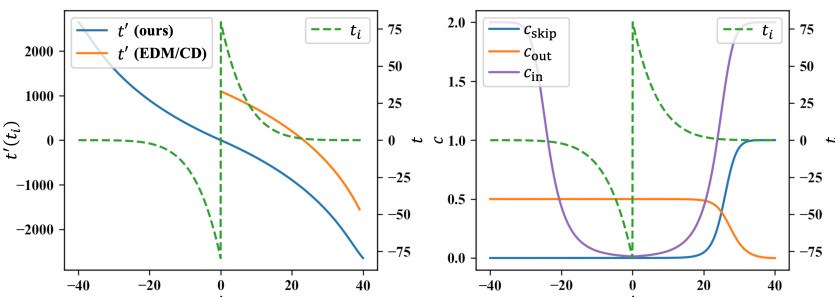

Figure 5: **Extension of EDM/CD model formulation for negative $t$ in IBCD student model.** $c_{\text{skip}}, c_{\text{out}}$, and $c_{\text{in}}$ represent when $c = c_b$ (the translation direction is $\mathcal{X}_A \to \mathcal{X}_B$).

- The interval $i = [-1, 0]$ uses the gradient at $-1$ (from domain A).
- The interval $i = [0, 1]$ uses the gradient at $0$ (from domain B).

In the backward direction (from domain B to domain A):

- The interval $i = [0, 1]$ uses the gradient at $1$ (from domain B).
- The interval $i = [-1, 0]$ uses the gradient at $0$ (from domain A).

Thus, due to the properties of the consistency function and the careful treatment of the non-differentiable point, the error bound of the consistency function remains $O((\Delta t)^p)$, consistent with the original CD approach. The validity of the Lipschitz condition and the equivalence of the local truncation error in the ODE solver ensure that the theorem holds true within the IBCD framework.

## C  IMPLEMENTATION DETAILS

**Model Architectures.** All models used in this study—the teacher $\phi$, student $\theta$, and fake DM $\psi$—employed the same model architecture as in EDM/CD (Karras et al., 2022; Song et al., 2023). The architecture configuration followed that of the LSUN-256 teacher EDM model introduced by Song et al. (2023). However, the student model was further modified with the model parametrization described in Appendix B, while the teacher and fake DM maintained the original EDM parametrization.

**Teacher Model Training.** The teacher model was trained using the EDM implementation and the LSUN-256 model training configuration provided by Song et al. (2023). The training setup included a log-normal schedule sampler and L2 loss, with a global batch size of 288, a learning rate of 1e-4, a dropout rate of 0.1, and an exponential moving average (EMA) of 0.9999. Mixed precision training was enabled, and weight decay was not applied. The teacher model was trained with class conditions on two types of AFHQ models (cat, dog, and wild) and CelebA-HQ models (female and male). The AFHQ and CelebA-HQ models were trained using their respective training sets from the AFHQ (Choi et al., 2020) and CelebA-HQ (Karras, 2018) datasets. Each model was trained for approximately 5 days, completing 800K steps on an NVIDIA A100 40GB eight-GPU setup.

**Implicit Bridge Consistency Distillation.** The discretization of DDIB trajectories is defined by extending the sampling discretization of EDM to satisfy Eq. (7):

$$t_i = \sigma_i = \begin{cases} \text{sign}(i)(\sigma_{\max}^{1/\rho} + \frac{|i|}{N-1}(\sigma_{\min}^{1/\rho} - \sigma_{\max}^{1/\rho}))^\rho & (N < i < N) \\ 0 & (i = \pm N) \end{cases} \tag{23}$$

where   $\text{sign}(x) = \begin{cases} +1 & (x \geq 0) \\ -1 & (x < 0) \end{cases}, \sigma_{\min} = 0.002, \; \sigma_{\max} = 80, \; \sigma_{\text{data}} = 0.5, \; N = 40, \; \rho = 7.0.$

For the distance function $d$ in each loss, $d_{\text{IBCD}}$ and $d_{\text{DMCD}}$ were based on LPIPS (Zhang et al., 2018), while $d_{\text{cycle}}$ used the L1 loss. The EMA parameter of the EMA model $\theta^-$ was 0.95, and an additional

EMA with a separate parameter 0.9999432189950708 was applied to the student model $\theta$ and used during inference. The global batch size was 256, with the student learning rate of 4e-5 and the fake DM learning rate of 1e-4. Dropout and weight decay were not used, and mixed precision learning was employed.

The ODE solver used was the 2nd order Huen solver (Ascher & Petzold, 1998), consistent with EDM/CD. The weight scheduler for the IBCD loss employed $\lambda(t) = 1$, while the DMCD loss used the weight scheduler $w_t$ as suggested in Yin et al. (2024). For the three tasks, Cat↔Dog, Wild↔Dog models were distilled using the AFHQ-256 teacher model and its corresponding training dataset. The Male↔Female models were distilled using the CelebA-HQ-256 teacher model and its training dataset.

The distillation process began with only the IBCD loss and transitioned to using the full loss set once the FID (Heusel et al., 2017) evaluation metrics stabilized (*i.e.* transition step). Distillation was conducted on the same NVIDIA A100 40GB eight hardware used for training the teacher model. Additional hyperparameters for each model and configuration are detailed in Tab. 4.

**Evaluation.** We followed the evaluation methodology and tasks outlined in EGSDE (Zhao et al., 2022). The publicly available evaluation code[2] was used without modification. Validation sets from the AFHQ and CelebA-HQ datasets were used as the evaluation datasets. All images in each validation set were translated using the respective task-specific models. For each image pair (source domain and translated target domain), PSNR, SSIM (Wang et al., 2004), LPIPS (Zhang et al., 2018), and CLIPScore (Hessel et al., 2021) were computed, and the average values across all pairs were reported.

FID (Heusel et al., 2017) was calculated using the `pytorch-fid`[3] library to measure the distance between the real target domain image distribution and the translated target image distribution. Following the methodology of Choi et al. (2020) and Zhao et al. (2022), images from the CelebA-HQ dataset were resized and normalized before FID calculation, while images for other tasks were evaluated without additional preprocessing. L2 distance measurement was not included in this evaluation.

Density-coverage (Naeem et al., 2020) was computed using `prdc-cli`[4] between the distribution of real target domain images and the distribution of images translated into the target domain, similar to the FID measurement. The measurement mode was `T4096` (features of the `fc2` layer of the ImageNet pre-trained VGG16 (Simonyan, 2014) model). The metric was computed for the entire dataset at once, without using mini-batches. Unlike FID, no specific transformation was applied for the CelebA-HQ dataset.

In the user study, twenty participants were recruited to perform a blinded, pairwise comparison between our method and each baseline. For each model and task, two identical source images were randomly selected, and their corresponding outputs—generated by each model—were presented side by side for evaluation. Participants were asked to make three separate selections for each comparison: one based on realism, one based on fidelity to the source image, and one based on overall preference. Results are reported as the average ratio of participants who preferred our model over each baseline for each criterion, averaged across all tasks.

**Baselines.** As baselines, we compare our method against several GAN-based methods, including CycleGAN (Zhu et al., 2017b), Self-Distance (Benaim & Wolf, 2017), GcGAN (Fu et al., 2019), LeSeSIM (Zheng et al., 2021), StarGAN v2 (Choi et al., 2020), and CUT (Park et al., 2020b). We also benchmark against diffusion model (DM)-based methods such as ILVR (Choi et al., 2021), SDEdit (Meng et al., 2022), EGSDE (Zhao et al., 2022), CycleDiffusion (Wu & De la Torre, 2023), and SDDM (Sun et al., 2023). Additionally, we compare our approach with UNSB (Kim et al., 2024a), a few-step Schrödinger bridge-based method, and the teacher DDIB (Su et al., 2023). Most of the comparison results are sourced from Zhao et al. (2022) except for the density-coverage, while the results for UNSB and DDIB are based on our re-implementations. Finally, we also include results from GPT-Image-1 (OpenAI, 2025), a multi-turn image generation and editing foundation model powered by GPT-4.1 (OpenAI, 2025), the latest iteration of OpenAI's multimodal large language model.

---

[2]https://github.com/ML-GSAI/EGSDE

[3]https://github.com/mseitzer/pytorch-fid

[4]https://github.com/Mahmood-Hussain/generative-evaluation-prdc

Table 4: **Specific hyperparameters employed by different models and configurations.**

| Model | Cat↔Dog | | Wild↔Dog | | Male↔Female | |
|---|---|---|---|---|---|---|
| Configuration | IBCD | IBCD$^\dagger$ | IBCD | IBCD$^\dagger$ | IBCD | IBCD$^\dagger$ |
| $\lambda_{\mathrm{DMCD}}$ | 1 | 0.18 | 0.2 | 0.2 | 0.02 | 0.02 |
| $\lambda_{\mathrm{cycle}}$ | 0.03 | 0.003 | 0.001 | 0.0003 | 0.00001 | 0.00003 |
| $g(\cdot)$ | 1 | | $\min(\log(\cdot) + 10)$ | | | |
| transition step | 200K | 200K | 200K | 200k | 500K | 500K |
| total distillation step | 210K | 230K | 210K | 230K | 510K | 520K |

**Reproductions.** To evaluate our method, we replicated UNSB and DDIB, two approaches that have not been previously evaluated on our benchmark datasets. For UNSB, we used the publicly available official code[5] for both training and inference, following the default configuration for the Horse→Zebra task and training the model for 400 epochs. During inference, we performed 5 steps. For DDIB, we implemented the method within our framework. Specifically, DDIB was executed by first solving the ODE backward from the source domain, then solving it forward again to the target domain using the EDM model trained for IBCD. The ODE solver was implemented in the same manner as the EDM sampler, utilizing the same sampling hyperparameters defined for EDM/IBCD. This setup ensured consistency in the evaluation and allowed for a direct comparison of performance across methods.

We also re-sampled the result from models (CUT, ILVR, SDEdit, EGSDE, CycleDiffusion) for which the density-coverage (Naeem et al., 2020), LPIPS (Zhang et al., 2018) and CLIPScore (Hessel et al., 2021), and user study metrics were not originally reported. These metrics were measured for these models using the method described above, and we included results in Tab. 2 and Tab. 6. The target models for this evaluation were limited to baseline models that met the following criteria: 1) Open-source code and checkpoints were available. 2) FID, PSNR, and SSIM values reported by the authors could be reproduced using the reported sampling strategy. This ensured that all metrics in Tab. 2 and Tab. 6 were measured on consistent samples.

The results for GPT-Image-1 were obtained by providing an input image to the `gpt-4.1-2025-04-14` model via the OpenAI API, instructing it to transform the image into the target domain using the `gpt-image-1-2025-04-23` tool. The input image was resized to 256×256 pixels, consistent with all other experiments, and the output image (originally 1024×1024) was downsampled to 256×256. The prompts used for generation are provided in the Tab. 5. This experiment incurred a total cost of approximately $100.

## D FURTHER EXPERIMENTAL RESULTS

### D.1 DISTILLATION ERROR IN VANILLA IBCD

Fig. 6 illustrates the distillation error that arises when using only vanilla IBCD loss on the synthetic toy dataset. When generating samples from pure noise to domain $B$ (Fig. 6 (a)) or translating samples from domain $A$ to domain $B$ (Fig. 6 (b)) using only IBCD loss, the translated results often fall in the low-density region of the target distribution. These translated points primarily originate from the source domain decision boundary, which is the boundary separating the partition in the source domain that should be mapped to two different target domain modes. Translation errors are more pronounced in longer neural jump paths, such as those involved in translations ($i = -N + 1 \rightarrow N - 1$), compared to shorter paths in generation ($i = 0 \rightarrow N - 1$).

### D.2 USER STUDY AND PERCEPTUAL METRIC EVALUATIONS

The results of the user study and perceptual metric evaluations are summarized in Tab. 6, which complement the main quantitative comparisons presented in Tab. 2. For the user study, we report the proportion of participants who preferred each baseline over our method (IBCD) in the pairwise comparisons. Values below 0.5 (blue) indicate that the majority preferred our method over the cor-

---

[5]https://github.com/cyclomon/UNSB

Table 5: **Prompts used in the GPT-Image-1 translation experiment for each task.** These prompts were provided to the `gpt-4.1-2025-04-14` model via the OpenAI API using the `gpt-image-1-2025-04-23` tool. Each prompt was designed to preserve pose, lighting, background, and composition while achieving realistic domain translation.

| Task | Prompt |
|---|---|
| Cat→Dog | Transform the image of the cat into a realistic dog, preserving the same pose, lighting, background, and overall composition. Ensure the dog appears natural and lifelike, matching the fur color, orientation, and proportions of the original cat as closely as possible. The final image should look photorealistic and faithful to the original scene, as if the dog were actually photographed in place of the cat. |
| Wild→Dog | Transform the image of the wild animal into a realistic dog, preserving the same pose, lighting, background, and overall composition. Ensure the dog appears natural and lifelike, matching the fur color, orientation, and proportions of the original animal as closely as possible. The final image should look photorealistic and faithful to the original scene, as if the dog were actually photographed in place of the wild animal. |
| Male→Female | Transform the image of the man into a realistic woman, preserving the same pose, lighting, background, and overall composition. Ensure the woman appears natural and lifelike, matching the skin tone, orientation, and proportions of the original man as closely as possible. The final image should look photorealistic and faithful to the original scene, as if the woman were actually photographed in place of the man. |

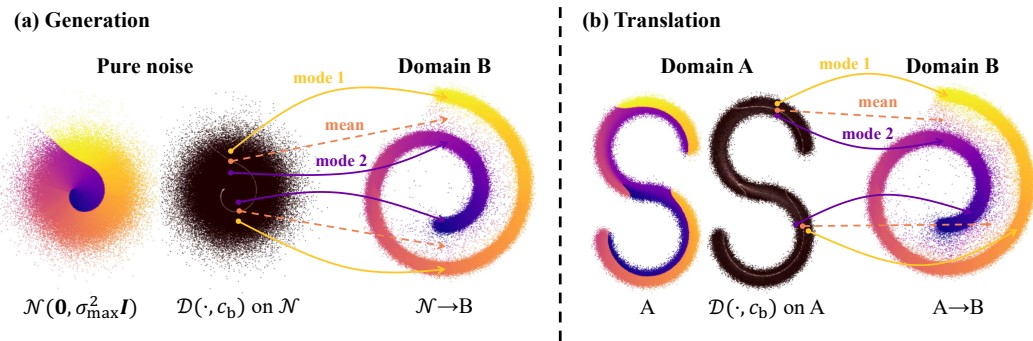

Figure 6: **Incorrect mapping to low-density regions due to distillation error.** (a) Generation and (b) translation results using vanilla IBCD. Samples with high distillation difficulty (*i.e.*, large distillation errors), which originate near the source domain decision boundary, tend to be mapped to low-probability regions in the target domain.

responding baseline. When multiple variants of a baseline exist, comparisons are made against the default version. Taken together with the results in Tab. 2, our approach demonstrates consistently strong performance across both automated metrics and human evaluations. We note that perceptual consistency may be further improved by replacing the current L1-based cycle loss with a perceptual loss formulation.

### D.3 EFFECT OF THE AUXILIARY LOSS WEIGHTS

Following the component ablation study of IBCD in the main text and Fig. 7, we further investigated the influence of auxiliary loss weights on translation outcomes. Specifically, we varied the weight of the DMCD loss $\lambda_{\text{DMCD}}$ and the cycle loss $\lambda_{\text{cycle}}$ in the Male→Female task (Fig. 8). During these experiments, the distillation difficulty adaptive weighting was not applied. The results aligned with expectations: as $\lambda_{\text{DMCD}}$ increases, the realism of the translation result improved, while increasing $\lambda_{\text{cycle}}$ enhanced the faithfulness of the translation. Thus, in the realism-faithfulness trade-off curve, the DMCD loss emphasizes realism, whereas the cycle loss emphasizes faithfulness.

Table 6: **User study and Perceptual Evaluation Results**. User study 1:1 win rates: the proportion of users preferring each baseline over our model (IBCD) in pairwise comparisons. Values below 0.5 indicate that the majority preferred our model. When two versions of a model exist, they are compared against the default version (without †).

| Method | NFE | User Study (1:1 win rate vs. IBCD) | | | Cat→Dog | | Wild→Dog | | Male→Female | |
| | | Reality ↑ | Faithfulness ↑ | Preference ↑ | LPIPS ↓ | CLIP ↑ | LPIPS ↓ | CLIP ↑ | LPIPS ↓ | CLIP ↑ |
|---|---|---|---|---|---|---|---|---|---|---|
| ILVR | 1000 | 0.25 | 0.49 | 0.25 | 0.454 | 72.65 | 0.486 | 67.22 | - | - |
| SDEdit | 1000 | 0.38 | **0.73** | 0.42 | 0.438 | 73.55 | 0.465 | 67.81 | 0.290 | 53.50 |
| EGSDE | 1000 | 0.37 | 0.28 | 0.43 | 0.433 | 73.98 | 0.467 | 67.34 | 0.284 | 53.12 |
| EGSDE† | 1200 | - | - | - | 0.497 | 73.53 | 0.526 | 66.25 | 0.343 | 51.11 |
| CycleDiffusion | 1000 | 0.25 | **0.62** | 0.25 | **0.381** | **74.77** | 0.417 | **67.84** | - | - |
| DDIB* (Teacher) | 160 | 0.42 | 0.23 | 0.42 | 0.475 | 73.65 | 0.492 | 67.05 | 0.326 | 56.20 |
| **IBCD (Ours)** | 1 | compared with each baseline | | | 0.384 | 74.00 | **0.404** | 66.86 | **0.261** | **56.57** |
| **IBCD† (Ours)** | 1 | - | - | - | 0.406 | 73.69 | 0.423 | 66.99 | **0.263** | **56.53** |

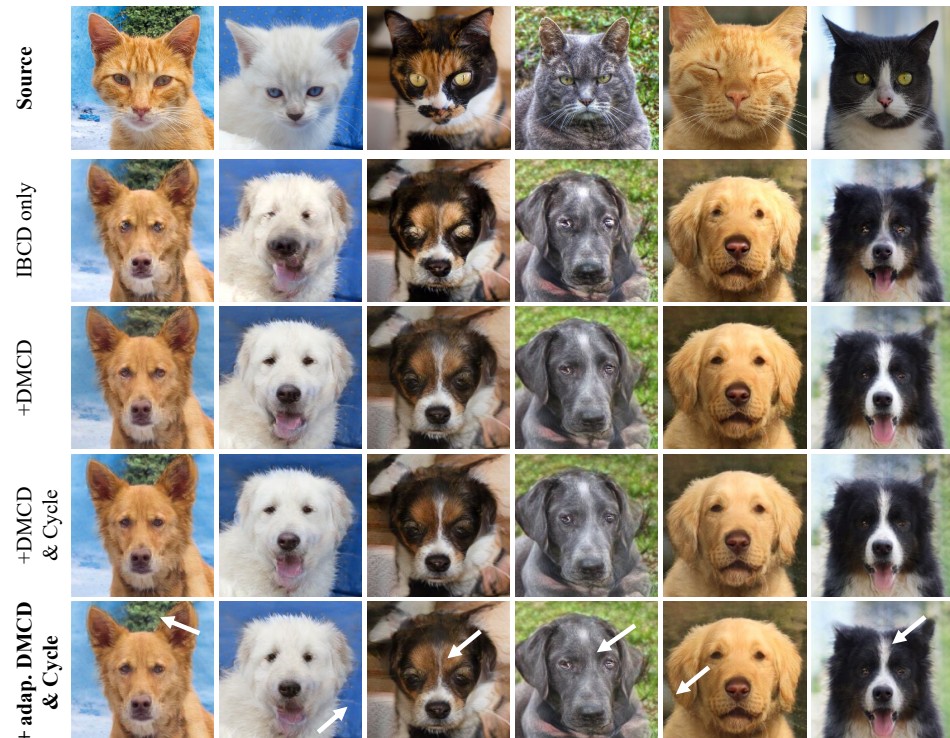

Figure 7: **Ablation study results on Cat→Dog task.** The DMCD loss improves the realism of the generated results compared to the vanilla IBCD. Additionally, the cycle translation loss and adaptive DMCD loss enhance source fidelity (as indicated by the arrows). These findings confirm that the components of IBCD work synergistically to achieve a better balance between realism and faithfulness.

## D.4 Approximated Distillation Difficulty in Image-to-image Translation

To explore the implications of the approximated distillation difficulty for real image-to-image translation tasks, we computed an expected approximated distillation difficulty $\mathbb{E}_{t \sim \mathcal{U}[-N+1, N-2]}[\hat{\mathcal{D}}(\mathbf{x}_t, c_{\text{FEMALE}})]$ for all trajectories generated with the DDIB teacher in the Male→Female task using the vanilla IBCD model. We then selected the trajectories with the top 10 and bottom 10 approximate distillation difficulties and performed Male→Female translation using the vanilla IBCD model for these trajectories, as shown in Fig. 9 without cherry-picking. The results indicate that the IBCD model struggles to effectively transform source images from trajectories with high approximate distillation difficulty into target images compared to those with low approximate distillation difficulty. Specifically, the translation results within the top 10 distillation

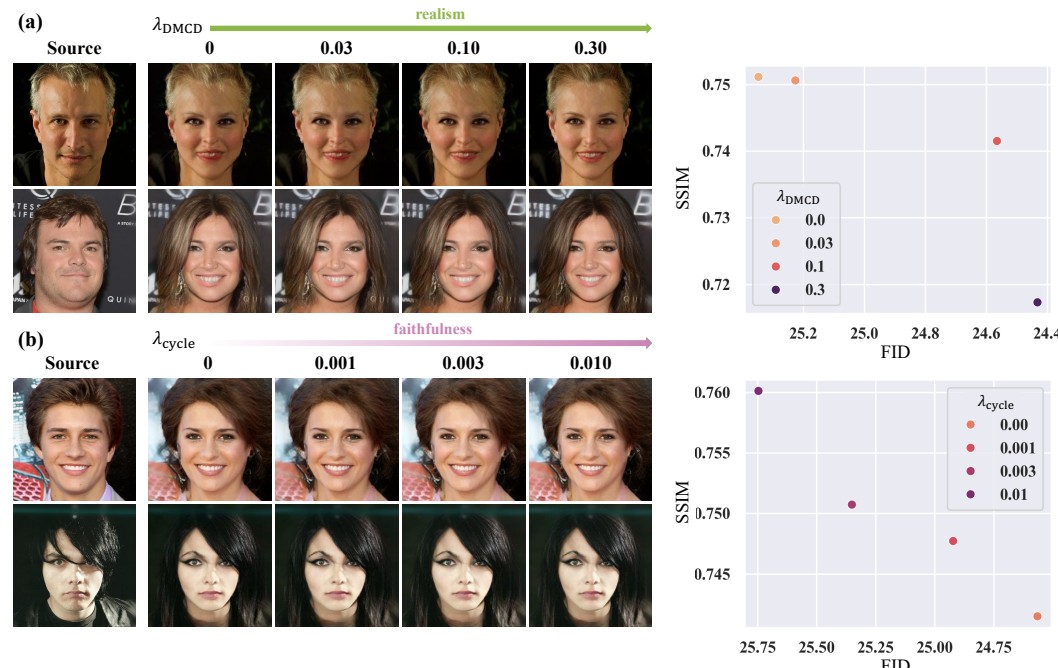

Figure 8: **Effect of the auxiliary loss weights ($\lambda_{\text{DMCD}}$, $\lambda_{\text{cycle}}$) for the Male→Female task.** In (a) $\lambda_{\text{cycle}}$ was set to 0, and in (b) $\lambda_{\text{DMCD}}$ was set to 0.10. Distillation difficulty adaptive waiting was not applied.

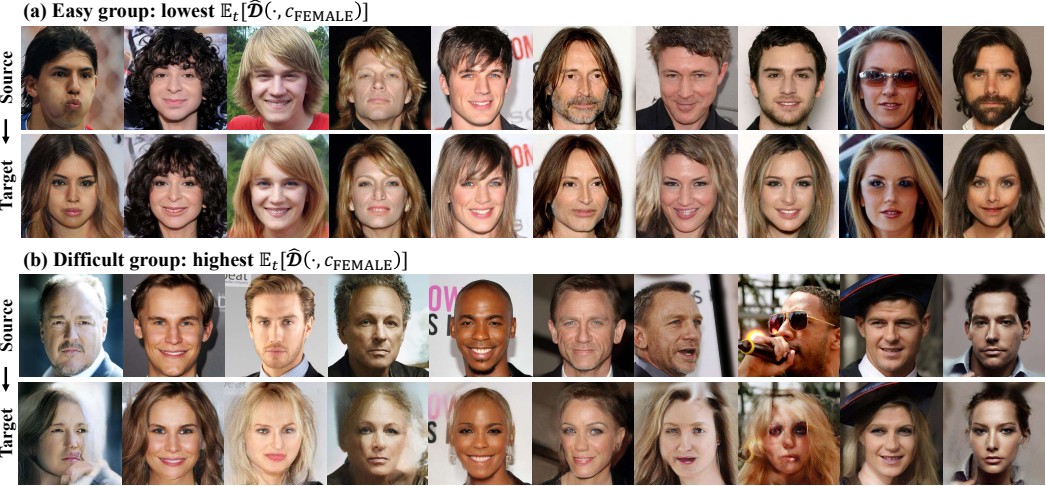

Figure 9: Relationship between self-assessed approximate distillation difficulty $\mathbb{E}_t[\hat{\mathcal{D}}(\cdot, c_{\text{FEMALE}})]$ and the translations performed in the Male→Female task.

difficulty groups exhibit relatively inferior image quality, highlighting the impact of distillation difficulty on translation performance.

## D.5 TRAINING STABILITY

Our framework does not rely on adversarial losses, which are known to be unstable and difficult to tune, giving it a clear advantage in training stability. To demonstrate this, we present the training loss curves in Fig. 10. As shown in the figure, both the consistency loss and the cycle loss steadily decrease throughout training, with a brief fluctuation only when the auxiliary losses are introduced.

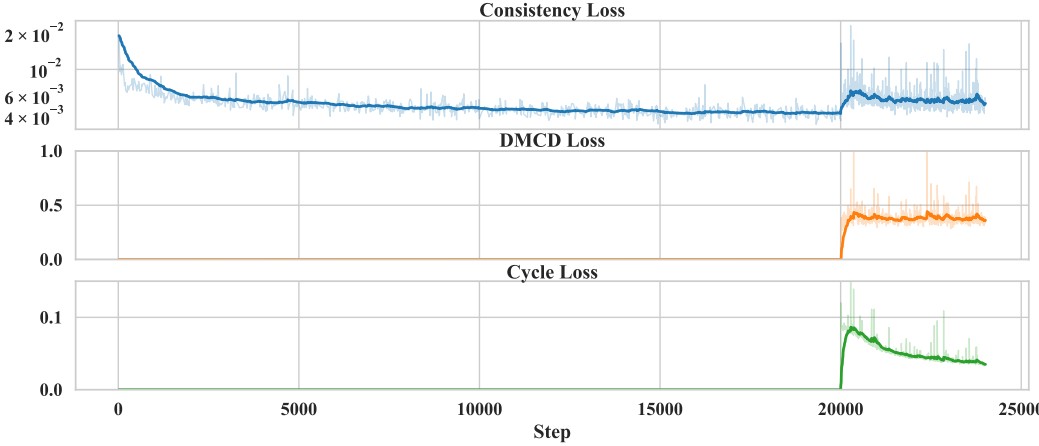

Figure 10: **Visualization of the training loss curve highlighting the stability of our framework on the Cat→Dog task.** DMCD loss and cycle loss are introduced starting at 20k training steps.

Table 7: **Quantitative comparison of model inference times and parameter sizes.**

| Method | Parameters [M] | NFE ↓ | Time [$s/img$] ↓ | Relative Time ↓ |
|---|---|---|---|---|
| StarGAN v2 (Choi et al., 2020) | 64.45 | 1 | 0.0052 | 0.43 |
| CUT (Park et al., 2020b) | 11.39 | 1 | 0.0070 | 0.58 |
| UNSB (Kim et al., 2024a) | 14.69 | 5 | 0.077 | 6.42 |
| ILVR (Choi et al., 2021) | 93.56 | 1000 | 13.40 | 1116.67 |
| SDEdit (Meng et al., 2022) | 93.56 | 1000 | 6.78 | 565.00 |
| EGSDE (Zhao et al., 2022) | 147.14 | 1000 | 15.89 | 1324.16 |
| CycleDiffusion (Wu & De la Torre, 2023) | 187.12 | 1000(+100) | 26.03 | 2169.17 |
| GPT-Image-1 (Foundation) (OpenAI, 2025) | $\gg 1$ | $\gg 1$ | 30.32 | 2526.67 |
| DDIB (Teacher) (Su et al., 2023) | 32.95 | 160 | 1.45 | 120.83 |
| **IBCD (Ours)** | 32.95 | 1 | 0.012 | 1 |

The DMCD loss also shows a stable plateau, as expected, since the fake score is computed by a jointly trained fake-score model. Overall, the loss trajectories indicate that our framework trains smoothly and remains stable throughout the process.

### D.6 MODEL INFERENCE EFFICIENCY

To reflect real-world constraints such as model size and inference algorithms, we conducted an inference speed comparison experiment. Instead of relying solely on NFE comparisons, we measured the actual inference time for a Cat→Dog task on a single NVIDIA GeForce RTX 4090 GPU (except for GPT-Image-1) with the batch size of 1. Tab. 7 presents the average inference time per image and the relative time for each methodology, and the number of model parameters. The results demonstrate that our methodology is computationally efficient in real-world sampling scenarios, while also using substantially fewer parameters than diffusion-based baseline methods.

### D.7 COMPARISON WITH OT AND SB BASELINES

We additionally conduct a quantitative comparison with optimal transport (OT)– and Schrödinger bridge (SB)–based baselines for completeness. The comparison is performed on the Cat→Dog image-to-image translation task, following the same evaluation protocol used in the main quantitative study. Specifically, we compare our method against NOT (Korotin et al., 2023), DIOTM (Choi et al., 2025), ASBM (Gushchin et al., 2024), DSBM (Shi et al., 2023), and Eg-EOT (Mokrov et al., 2024). Since these methods do not natively support our task, we retrained all baselines using their highest supported resolution settings, matched to our dataset and resolution.

Tab. 8 and Fig. 11 show that IBCD achieves superior performance compared to both OT and SB baselines. Notably, all SB/OT methods fail to scale to our higher-resolution setting under their de-

Table 8: **Quantitative comparison of unpaired image-to-image translation tasks with OT- and SB-based baselines.**

| Method | Parameters [M] | NFE↓ | FID↓ | PSNR↑ | SSIM↑ | Density↑ | Coverage↑ |
|---|---|---|---|---|---|---|---|
| | | | Cat→Dog | | | | |
| NOT (Korotin et al., 2023) | 9.72 | 1 | 161.54 | 15.12 | 0.566 | 0.531 | 0.072 |
| DIOTM (Choi et al., 2025) | 39.65 | 1 | 75.70 | 12.03 | 0.363 | 1.215 | 0.590 |
| ASBM (Gushchin et al., 2024) | 79.58 | 4 | 91.40 | 17.71 | 0.463 | 0.871 | 0.478 |
| DSBM (Shi et al., 2023) | 131.02 | 100 | 100.08 | **21.24** | 0.532 | 0.750 | 0.396 |
| Eg-EOT (Mokrov et al., 2024) | 26.21 | 100 | 53.29 | 15.93 | 0.349 | 1.085 | 0.626 |
| DDIB (Teacher) (Su et al., 2023) | 32.95 | 160 | 38.91 | 17.58 | 0.588 | 1.528 | 0.934 |
| **IBCD (Ours)** | 32.95 | 1 | **47.44** | 19.50 | **0.701** | **1.412** | **0.940** |
| **IBCD† (Ours)** | 32.95 | 1 | **44.77** | 18.04 | **0.663** | **1.542** | **0.935** |

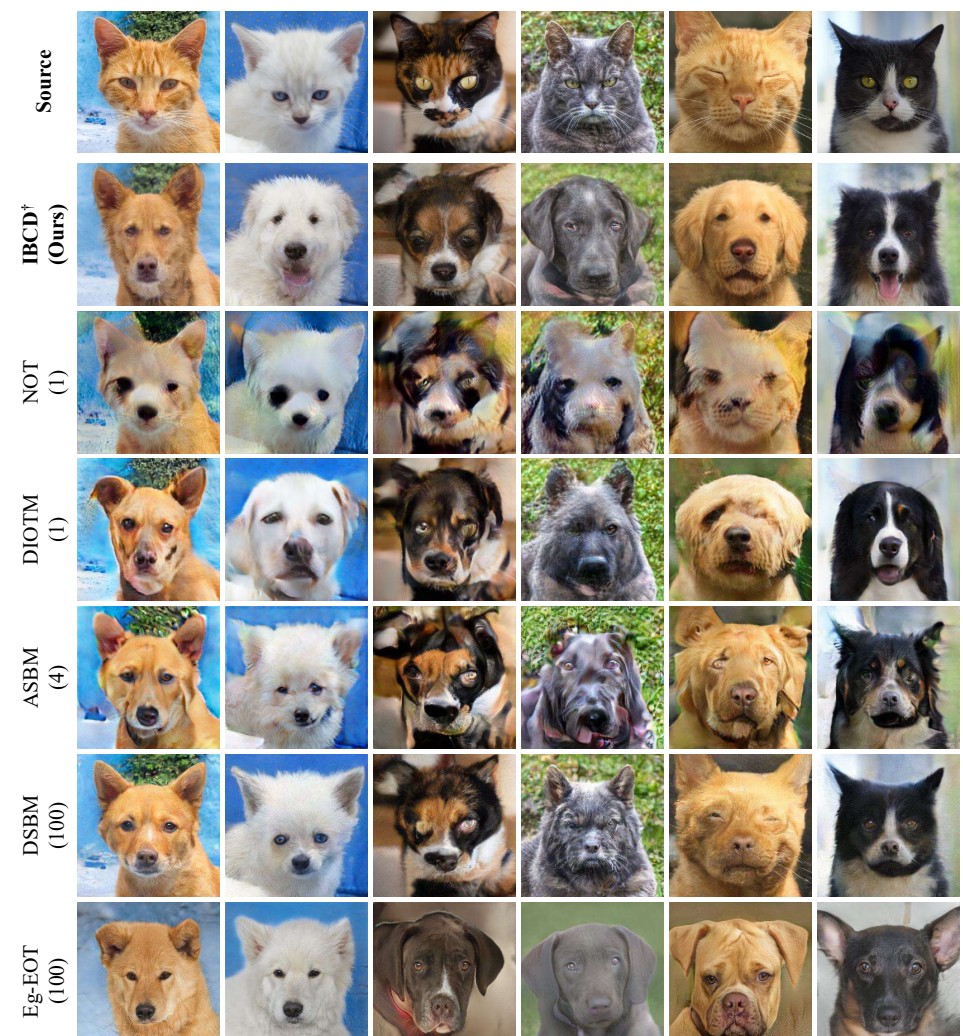

Figure 11: **Qualitative comparison of unpaired image-to-image translation tasks with OT- and SB-based baselines**. The numbers in parentheses represent inference NFE.

fault configurations, highlighting the curse of dimensionality inherent to high-dimensional OT and SB formulations (with the exception of Eg-EOT, which operates in the StyleGAN2-ADA latent space (Karras et al., 2020)). DSBM also reflects this limitation: its higher source faithfulness results from failing to produce meaningful target-domain translations at higher resolutions, leading to minimal changes and thus significantly worse FID, Density, and Coverage metrics.

### D.8 FAILURE CASES

IBCD occasionally produces failure cases as illustrated in Fig. 12. The primary failures can be attributed to incomplete translations (Fig. 12(a)) and incorrect cycle translations (Fig. 12(b)), which are likely due to distillation errors and the side effects of auxiliary losses. Distillation errors from the CD, in particular, appear to be the primary reason. The DMCD and cycle translation loss can also contribute to these issues, with the former leading to incorrect cycle translations and the latter to incomplete translations. Minimizing distillation errors through improved distillation methods and advanced weighting strategies for auxiliary losses might address this issue.

### D.9 BIDIRECTIONAL TRANSLATIONS

To evaluate IBCD's bidirectional translation capabilities, we compared it to baseline methods through two tasks: *opposite translation* and *cycle translation*. Opposite translation involves reversing the main translation task (Dog→Cat, Dog→Wild, Female→Male), while cycle translation involves performing the reverse task after the main translation (Cat→Dog→Cat, Wild→Dog→Wild, Male→Female→Male). To ensure a fair comparison of bidirectional performance, we used the same model and sampling hyperparameters for each domain pair (Cat↔Dog, Wild↔Dog, Male↔Female) in both opposite and cycle translation tasks.

Given the limited number of models capable of bidirectional translation, we selected StarGAN v2 (Choi et al., 2020), CycleDiffusion (Wu & De la Torre, 2023), and DDIB (teacher) (Su et al., 2023) as baselines. We measured FID for the final target domain for the cycle translation task. It's worth noting that StarGAN v2's inference process differs from its main translation task (Tab. 2) performed by Zhao et al. (2022) for a better fair comparison. It inputs the same source image as both the source and reference images, enabling it to achieve both high realism and faithfulness.

Tab. 9 and Fig. 13, 14 demonstrate that our model also excels in reverse and cycle translation tasks, exhibiting the best performance and high efficiency. This further supports its strong bidirectional translation capabilities.

### D.10 UNPAIRED MRI CONTRAST TRANSLATION

We conducted experiments on the BraTS2021 dataset (Baid et al., 2021) to evaluate our model on the brain MRI contrast translation task, demonstrating its applicability in the medical imaging domain. We used T1- and T2-weighted brain MRI scans from the dataset (excluding other contrasts) and performed unpaired translation from T1- to T2-weighted images at a resolution of 256×256. Although the dataset provides paired images, we utilized them in an unpaired setting during training. The dataset was split into 1,126 volumes (174,530 images) for training and 10 volumes (1,550 images) for validation. Quantitative and qualitative comparisons with existing baselines (Tab.15 and Fig.10) demonstrate the effectiveness of our method, highlighting its potential for practical deployment in clinical workflows.

Table 10: Quantitative comparison of unpaired MRI contrast translation tasks.

| Method | compared with G.T. | |
| --- | --- | --- |
| | PSNR ↑ | SSIM ↑ |
| SDEdit | 29.85 | 0.849 |
| EGSDE | 23.34 | 0.842 |
| CycleDiffusion | 30.52 | 0.825 |
| GPT-Image-1 (Foundation) | not working | |
| DDIB* (Teacher) | 30.24 | 0.825 |
| **IBCD (Ours)** | **33.28** | **0.855** |

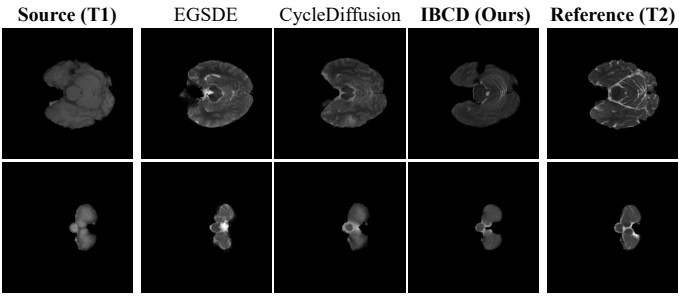

Figure 15: Qualitative comparison of unpaired MRI contrast translation tasks.

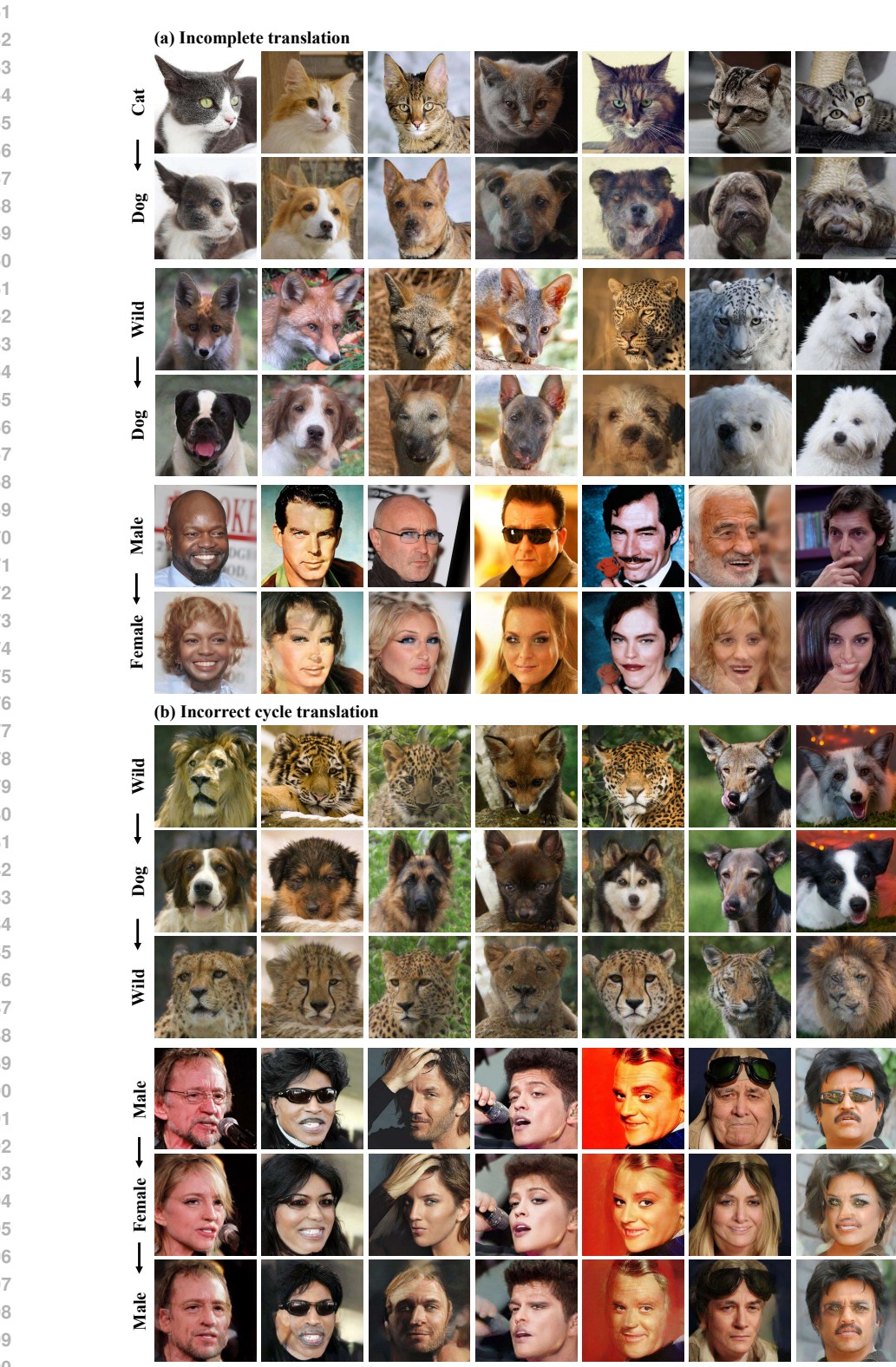

Figure 12: Example of failure cases, which are (a) incomplete translation and (b) incorrect cycle translation.

Table 9: **Quantitative comparison of unpaired image-to-image translation tasks (opposite & cycle translation)**. The opposition task used the same model and inference hyperparameters as the main direction task using bi-directionality.

| Method | NFE ↓ | FID ↓ | PSNR ↑ | SSIM ↑ | Density ↑ | Coverage ↑ |
|---|---|---|---|---|---|---|
| **Dog→Cat** | | | | | | |
| StarGAN v2 (Choi et al., 2020) | 1 | 37.73 | 16.02 | 0.399 | 1.336 | 0.778 |
| CycleDiffusion (Wu & De la Torre, 2023) | 1000(+100) | 40.45 | 17.83 | 0.493 | 1.064 | 0.774 |
| DDIB (Teacher) (Su et al., 2023) | 160 | 30.28 | 17.15 | 0.597 | 2.071 | 0.902 |
| **IBCD (Ours)** | 1 | 28.99 | **19.10** | **0.695** | 1.699 | 0.894 |
| **IBCD† (Ours)** | 1 | **28.41** | 17.40 | 0.653 | **2.112** | **0.920** |
| **Dog→Wild** | | | | | | |
| StarGAN v2 (Choi et al., 2020) | 1 | 49.35 | 16.17 | 0.386 | 0.772 | 0.478 |
| CycleDiffusion (Wu & De la Torre, 2023) | 1000(+100) | 27.01 | 16.99 | 0.421 | 0.816 | 0.752 |
| DDIB (Teacher) (Su et al., 2023) | 160 | 13.20 | 16.80 | 0.583 | 1.202 | 0.760 |
| **IBCD (Ours)** | 1 | 18.79 | **17.56** | **0.671** | 0.900 | **0.830** |
| **IBCD† (Ours)** | 1 | **16.67** | 16.22 | 0.646 | **1.058** | 0.814 |
| **Female→Male** | | | | | | |
| StarGAN v2 (Choi et al., 2020) | 1 | 59.56 | 15.75 | 0.465 | 1.145 | 0.587 |
| DDIB (Teacher) (Su et al., 2023) | 160 | 26.98 | 18.74 | 0.668 | 1.154 | 0.858 |
| **IBCD (Ours)** | 1 | **31.28** | 19.93 | **0.733** | 1.300 | 0.808 |
| **IBCD† (Ours)** | 1 | 31.49 | **19.51** | 0.726 | **1.311** | **0.809** |
| **Cat→Dog→Cat** | | | | | | |
| StarGAN v2 (Choi et al., 2020) | 1 | 30.53 | 16.30 | 0.382 | 1.717 | 0.890 |
| CycleDiffusion (Wu & De la Torre, 2023) | 1000(+100) | 39.59 | 19.01 | 0.434 | 0.731 | 0.676 |
| DDIB (Teacher) (Su et al., 2023) | 160 | 16.56 | 25.88 | 0.804 | 1.330 | 0.990 |
| **IBCD (Ours)** | 1 | **22.42** | **22.35** | **0.767** | 1.322 | **0.992** |
| **IBCD† (Ours)** | 1 | 24.03 | 20.28 | 0.724 | **1.749** | 0.988 |
| **Wild→Dog→Wild** | | | | | | |
| StarGAN v2 (Choi et al., 2020) | 1 | 37.76 | 15.30 | 0.285 | 1.102 | 0.566 |
| CycleDiffusion (Wu & De la Torre, 2023) | 1000(+100) | 19.43 | 16.39 | 0.281 | 0.649 | 0.616 |
| DDIB (Teacher) (Su et al., 2023) | 160 | 6.75 | 26.08 | 0.803 | 1.118 | 0.974 |
| **IBCD (Ours)** | 1 | **9.89** | **20.56** | **0.739** | 1.118 | **0.972** |
| **IBCD† (Ours)** | 1 | 10.66 | 18.80 | 0.693 | **1.259** | 0.968 |
| **Male→Female→Male** | | | | | | |
| StarGAN v2 (Choi et al., 2020) | 1 | 57.80 | 15.39 | 0.502 | 1.634 | 0.728 |
| DDIB (Teacher) (Su et al., 2023) | 160 | 28.29 | 27.70 | 0.853 | 0.821 | 0.993 |
| **IBCD (Ours)** | 1 | **39.84** | **22.22** | **0.790** | **1.341** | 0.979 |
| **IBCD† (Ours)** | 1 | 39.96 | 21.85 | 0.783 | 1.332 | **0.984** |

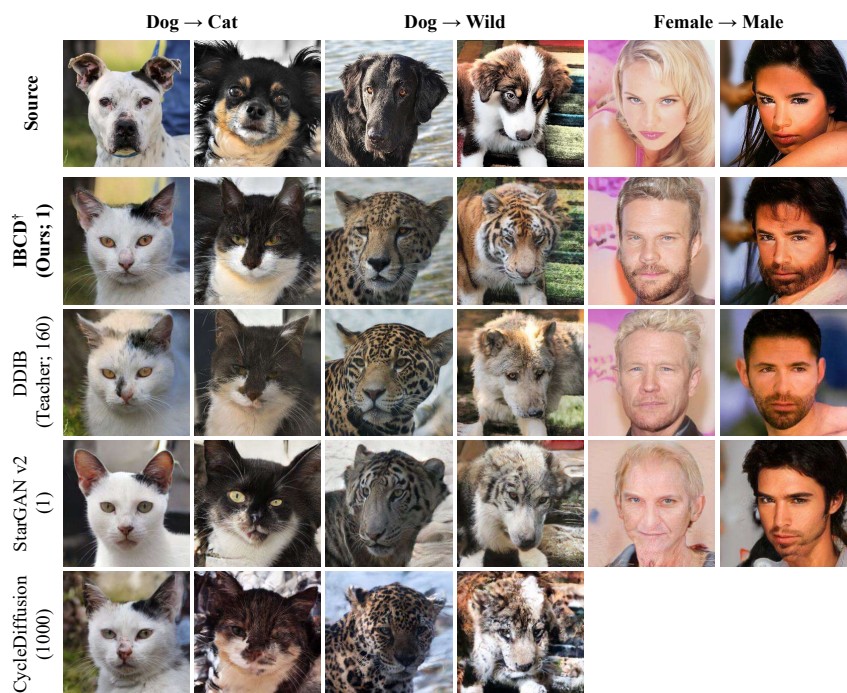

Figure 13: **Qualitative comparison of unpaired image-to-image translation tasks (opposite translation)**. Compared to other baselines, our model achieves more realistic and source-faithful translations in a single step. The numbers in parentheses represent inference NFE.

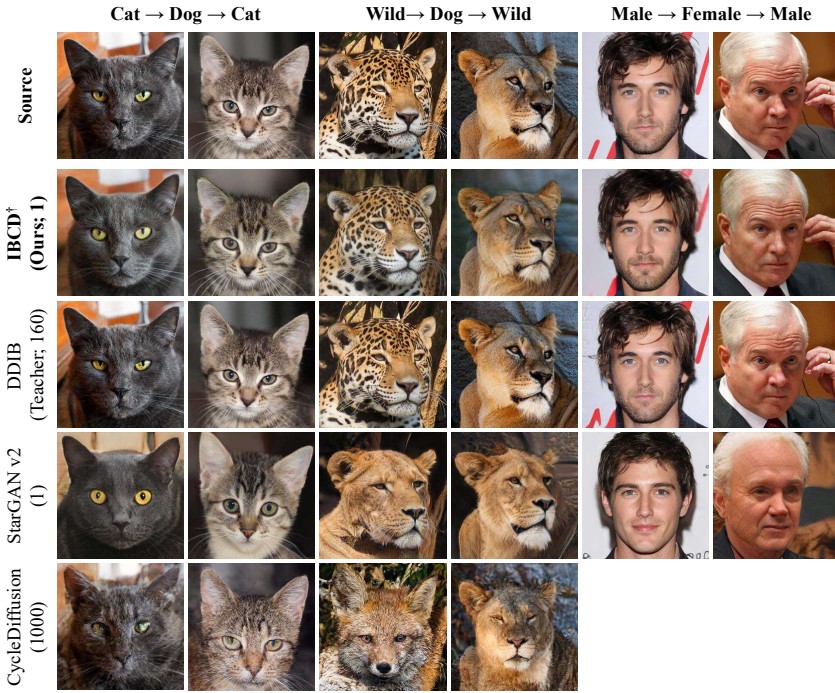

Figure 14: **Qualitative comparison of unpaired image-to-image translation tasks (cycle translation)**. Compared to other baselines, our model achieves consistent cycle translations in a single step. The numbers in parentheses represent inference NFE.

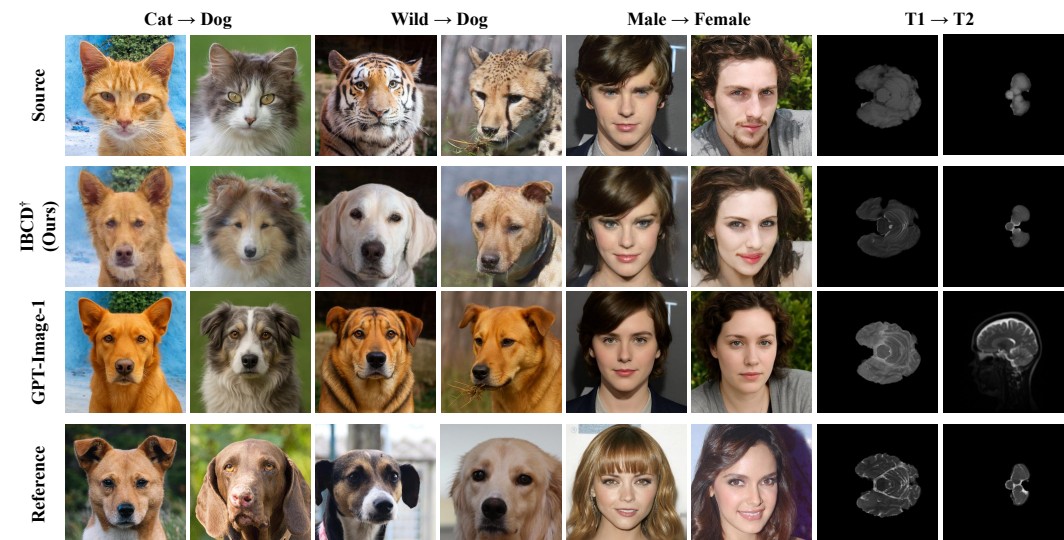

Figure 16: Limitations of foundational models in domain-specific translation.

## D.11  LIMITATIONS OF FOUNDATIONAL MODELS IN DOMAIN-SPECIFIC TRANSLATION

The rise of foundational diffusion-based image editing methods (Parmar et al., 2023; Hertz et al., 2023; 2022), particularly those coupled with large language models, has enabled intuitive zero-shot multi-turn editing (OpenAI, 2025; Team, 2025). Despite their appeal, unpaired I2I translation remains essential, especially in domains where paired data is scarce, such as medical and scientific imaging (Kaji & Kida, 2019; Chen et al., 2023).

These tasks demand specific, accurate knowledge of both source and target domains, along with fine-grained detail preservation and low-latency performance. Foundational models, while flexible, lack such specialization. As shown in Fig.16, GPT-Image-1 (OpenAI, 2025) often generates outputs that deviate significantly from the target distribution under zero-shot conditions, producing results that may appear unrealistic, incomplete, or exaggerated. This tendency is especially pronounced in medical scenarios like MRI translation, where the model frequently fails to produce clinically meaningful outputs. These limitations are further supported by the quantitative results in Tab. 2 and Tab. 15.

We do not claim our method outperforms foundational models across the board. Instead, we argue that unpaired I2I approaches remain indispensable in scenarios where zero-shot methods fall short due to their lack of domain-specific adaptation.

## D.12  MORE QUALITATIVE RESULTS

In this section, we present additional qualitative results obtained through cycle translation tasks (Cat→Dog→Cat, Wild→Dog→Wild, Male→Female→Male). The results of the Cat↔Dog, Wild↔Dog, and Male↔Female model are illustrated in Fig. 17, 18, 19. These results highlight our model's one-way and bidirectional translation capabilities.

**Cat** ⟶ **Dog** ⟶ **Cat**        **Dog** ⟶ **Cat** ⟶ **Dog**

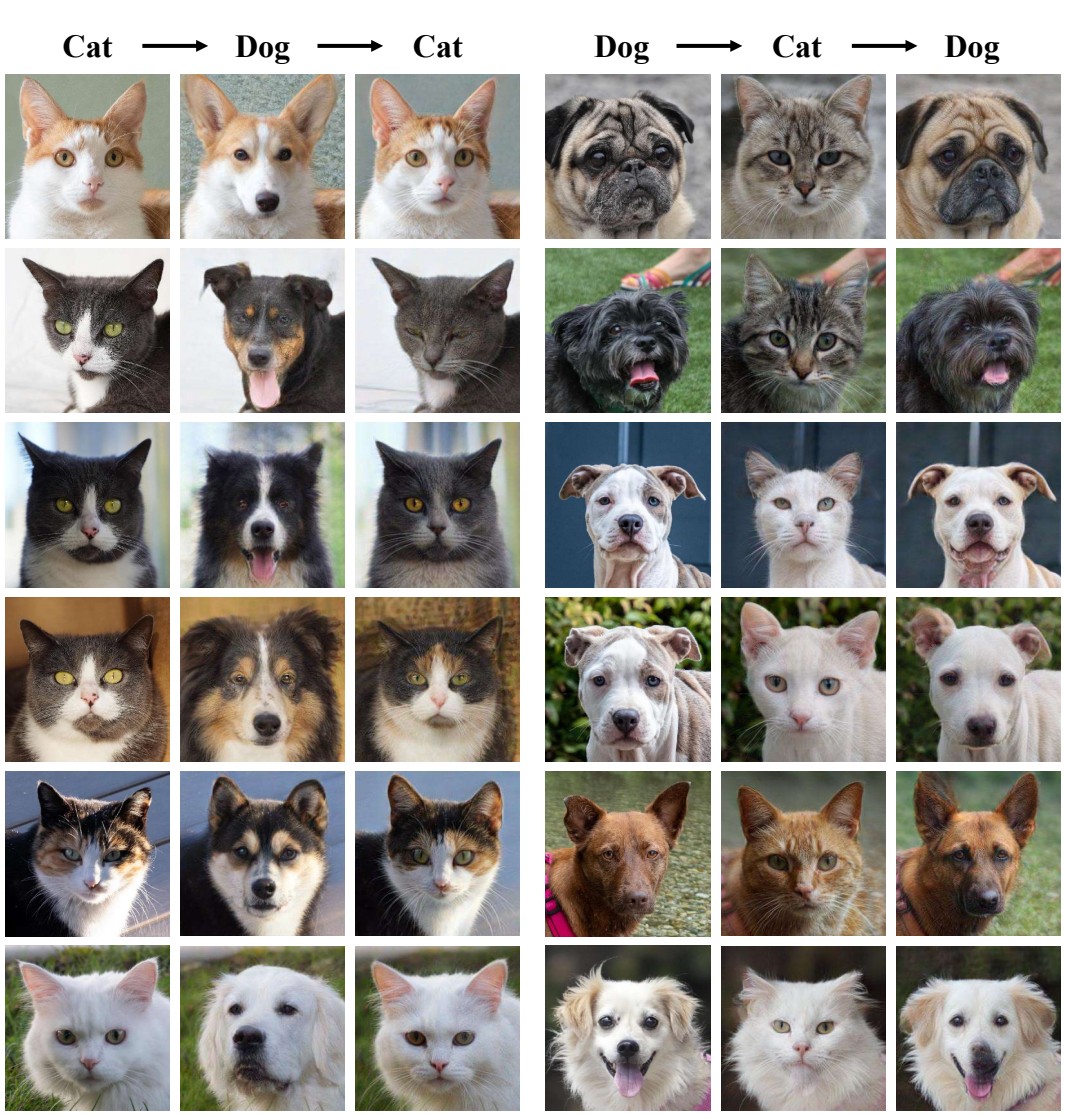

Figure 17: Result of the bi-directional cycle translation with a single model for the Cat↔Dog task (IBCD[†]).

**Wild** ⟶ **Dog** ⟶ **Wild**  **Dog** ⟶ **Wild** ⟶ **Dog**

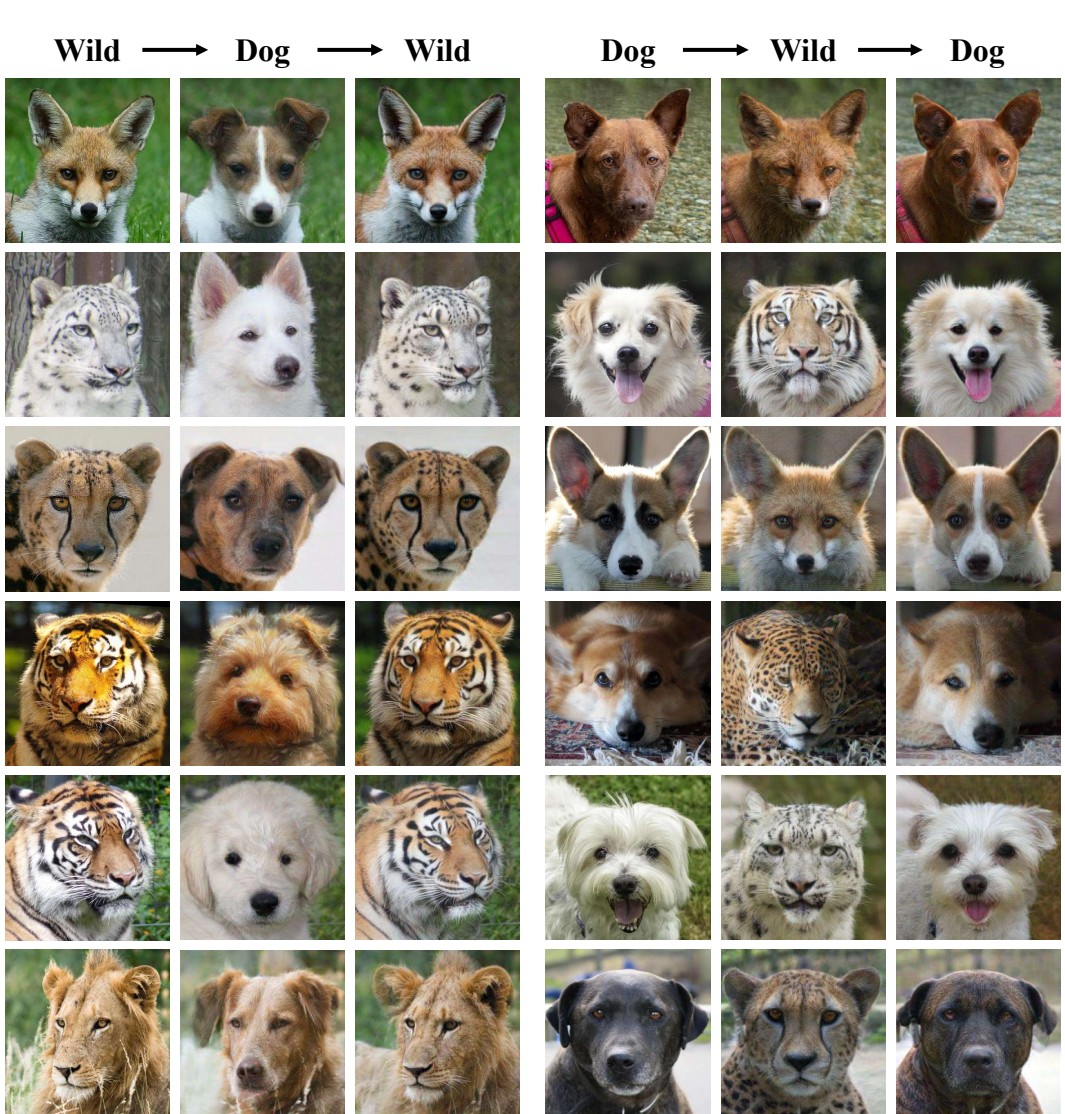

Figure 18: Result of the bi-directional cycle translation with a single model for the Wild↔Dog task (IBCD[†]).

**Male ⟶ Female ⟶ Male**   **Female ⟶ Male ⟶ Female**

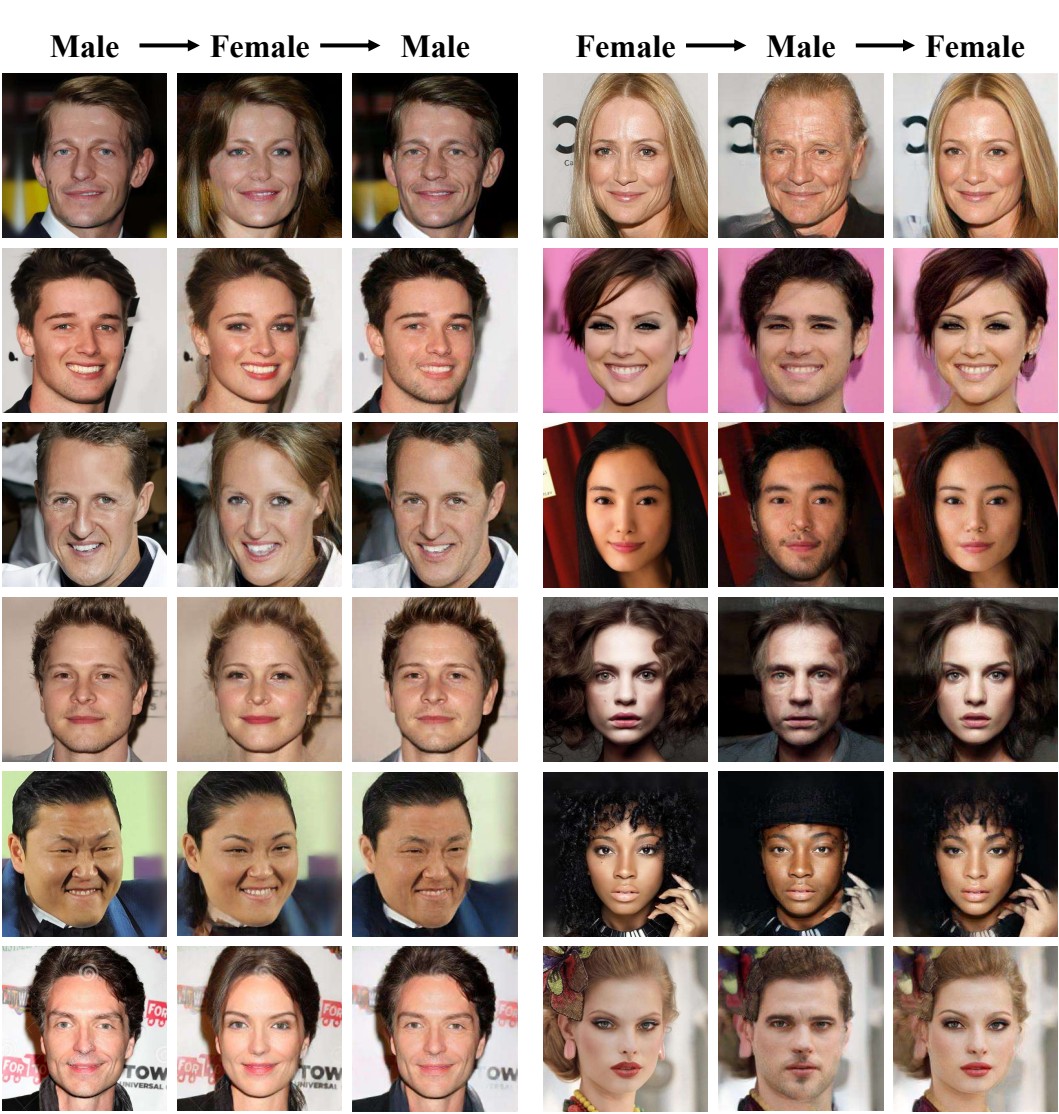

Figure 19: Result of the bi-directional cycle translation with a single model for the Male↔Female task (IBCD$^{\dagger}$).

