# OpenReview forum: "Single-Step Bidirectional Unpaired Image Translation Using Implicit Bridge Consistency Distillation"
_ICLR.cc/2026/Conference — Submitted to ICLR 2026_

### Official Review · Reviewer_fmc9 · 2025-10-16

**Soundness:** 3
**Presentation:** 3
**Contribution:** 3
**Rating:** 6
**Confidence:** 3

**Summary:**

The work proposed a single-step bidirectional unpaired image-to-image translation model using the combination of consistency distillation and distribution matching distillation for DDIB teacher model. Key contributions of the work include the formulation of consistency distillation for PF-ODE trajectories obtained with DDIB model and the extension of DMD framework for the proposed consistency distillation. The evaluation of the proposed IBCD model was done using cat-to-dog, wild-to-dog, and male-to-female unpaired image-to-image translation problems. IBCD is compared with adversarial methods, including CUT and UNSB, and diffusion methods, including EGSDE, CycleDiffusion, SDDM, SDEdit, and the teacher model DDIB. These image-to-image translation models are evaluated using FID, density and coverage for image realism, inference time and NFE for inference efficiency, and PSNR and SSIM for input-output similarity. The results of IBCD show its advantages against the teacher model DDIB in terms of the inference efficiency and input-output similarity, and other adversarial and diffusion models in terms of image realism.

**Strengths:**

1) Surprisingly, according to Table 2 and Table 7, the proposed method has the inference time, which is even less than for GANs, while having better realism and input-output similarity metrics. It makes diffusion models much closer for applications in unpaired image-to-image translation problems.
2) The novel adaptive DMCD loss greatly improves the input-output similarity, as demonstrated by Table 3 and Figure 7.
3) The extension evaluation of IBCD with different image-to-image translation metrics in Table 2 is also supported by user study and perceptual measures in Table 6. The method is also compared with LLM-based GPT-Image-1 model, which shows existing limitations in domain specific problems of foundational large models in zero-shot editing.
4) Figure 8 studies the effect of realism-similarity trade-off for the proposed model by balancing between DMCD and cycle-consistency losses

**Weaknesses:**

1) According to Table 4, the distillation of DDIB model requires more than 200k steps for the training IBCD. It remains unclear how fast and stable the distillation with the IBCD is performed compared to the training of other unpaired image-to-image translation models.
2) As pointed in many prior works on image-to-image translation problems [1, 2, 3], diversity remains an important characteristic of image-to-image translation models for multimodal pairs of domains. Even though authors provide standard deviations of the image quality metrics, the study lacks of diversity evaluation for the proposed method.
3) As pointed out in [1], optimization of cycle-consistency losses struggles for pairs of image domains, where there is a big complexity difference and bijection assumption does not hold, for example, for the sketch-to-image problem. The study lacks of such pairs for image domains in the evaluation protocol.
4) The work lacks of explanation about the inference efficiency of IBCD compared with GAN-based image-to-image translation models such as number of training parameters and model sizes, which seems to be unexpected.

[1]. Augmented CycleGAN: Learning Many-to-Many Mappings from Unpaired Data. ICML-2018.
[2]. Multimodal Unsupervised Image-to-Image Translation. ECCV-2018.
[3]. StarGAN v2: Diverse Image Synthesis for Multiple Domains. CVPR-2020.

**Questions:**

1) Can you comment on the training time and stability of the distillation for IBCD compared with baselines?
2) Can you quantify the diversity of IBCD model and baselines, for example, following the MUNIT approach [2]?
3) Can you comment on the number of training parameters and model size of IBCD compared to other baselines? The result of IBCD, which is 5 times faster than StarGAN-v2, looks impressive and I would like to understand this difference.
4) Can you comment on the applicability and performance of IBCD on the pairs of image domains, where the image-to-image translation is multimodal by the nature and bijection assumption does not hold? For example, for the problem of sketch-to-image translation.
5) Can you comment on the choice of the distance function $d$ and how it affects the results? For example, some methods employ $L_2$ distance instead of LPIPS in consistency loss [4].
6) DMD loss seem to be applied in previous works for diffusion unpaired image-to-image translation models. I suggest authors discuss the relation of their implementation of DMD loss and its implementation in the paper [5].

[4] Consistency Trajectory Models: Learning Probability Flow ODE Trajectory of Diffusion. ICLR-2024.
[5] Regularized Distribution Matching Distillation for One-step Unpaired Image-to-Image Translation. Structured Probabilistic Inference & Generative Modeling workshop of ICML 2024.

---

> ### Author Response · Authors · 2025-11-24
>
> **W1/Q1. Given that IBCD distillation for DDIB takes over 200k steps, how do its training time and stability compare with those of the baselines?**
>
> For the EDM teacher model, we trained it for 800K steps (5 days), followed by distillation of the student model for an additional 210K steps (3 days). Importantly, the total number of training steps for the teacher and student combined (1.1M steps) is comparable to the diffusion-based baselines, which were trained for 1M steps while incurring over 1000× higher inference cost. Moreover, the baselines rely on models that are roughly three times larger than ours in parameter count (see answer to Q3).
>
> We also include additional analyses and visualizations demonstrating stable optimization behavior and clearer training dynamics (Appendix D.5 and Fig. 10). The training curves for both Stage 1 and Stage 2 exhibit stable decreases and plateaus, mirroring the well-behaved dynamics typically observed in diffusion model training.
>
> **W2/Q2. Even though authors provide standard deviations of the image quality metrics, the study lacks of diversity evaluation for the proposed method. Can you quantify the diversity of IBCD model and baselines, for example, following the MUNIT approach [2]?**
>
> The conditional inception score (CIS) suggested by the reviewer measures the diversity of outputs for the same input. However, our framework does not incorporate a mechanism for producing multiple diverse outputs from a single source image; the model is designed to generate a single deterministic one-step mapping. Therefore, CIS is not applicable to our setting, as multimodal generation is outside the scope of our objective.
>
> That said, when diversity is evaluated at the distribution level, our method demonstrates strong performance compared to the baselines. In addition to achieving the best FID, which jointly reflects sample quality and diversity, our method also outperforms all baselines on Density and Coverage [6], which separately measure sample fidelity and sample diversity. These results indicate that our one-step distillation procedure preserves both the quality and diversity of the target distribution.
>
> **W3/Q4. Given that cycle-consistency struggles for non-bijective, multimodal tasks like sketch-to-image [1], which are not included in the evaluation, can you comment on IBCD’s applicability and expected performance in such settings?**
>
> Although our objective is formulated under the bijection assumption, our framework can be naturally extended to settings where this assumption does not hold. This is possible because our distillation target is the DDIB trajectories. DDIB produces one-to-one correspondences between domains even when the underlying translation is multimodal; in such cases, it effectively selects one plausible mapping consistent with the entropy-regularized objective. The consistency loss then guides the student to follow this mapping. As a result, both (1) the mapping behavior and (2) the target distribution-matching effect primarily arise from the consistency loss.
>
> This contrasts with the scenario described in [1], where cycle consistency is the main mechanism enforcing the mapping, making the method sensitive to violations of bijectivity. In our framework, cycle consistency serves only as an auxiliary refinement step rather than the component that defines the mapping itself. For this reason, our method remains applicable—and can be extended—even when the translation is non-bijective or multimodal.

---

> > ### Author Response · Authors · 2025-11-24
> >
> > **W4/Q3. The work lacks an explanation about the inference efficiency of IBCD compared with GAN-based image-to-image translation models, such as the number of training parameters and model sizes, which seems to be unexpected.**
> >
> > Following the suggestion from Reviewer Dtkz regarding inference-time evaluation, we conducted a more precise measurement by using a fixed batch size of 1 and removing warm-up and other overhead that do not reflect the actual forward computation. The updated results, including model parameter sizes, are reported in Tab. 7 of the revised manuscript. These results show that our method remains substantially faster than the diffusion baseline. Although it is still slower than GAN-based models, the difference is moderate and justified by the significant performance improvements.  A summary of the updated Tab. 7  is provided below.
> >
> > *Table A: Quantitative comparison of model inference times and parameter sizes.*
> > | Model          | Parameters [M] |    NFE ↓   | Time [s/img] ↓ | Relative Time ↓ |
> > |----------------|:--------------:|:----------:|:--------------:|:---------------:|
> > | StarGAN v2     |      64.45     |      1     |     0.0052     |       0.43      |
> > | CUT            |      11.39     |      1     |     0.0070     |       0.58      |
> > | UNSB           |      14.69     |      5     |      0.077     |       6.42      |
> > | ILVR           |      93.56     |    1000    |      13.40     |     1116.67     |
> > | SDEdit         |      93.56     |    1000    |      6.78      |      565.00     |
> > | EGSDE          |     147.14     |    1000    |      15.89     |     1324.16     |
> > | CycleDiffusion |     187.12     | 1000(+100) |      26.03     |     2169.17     |
> > | GPT-Image-1    |       >>1      |     >>1    |      30.32     |     2526.67     |
> > | DDIB (Teacher) |      32.95     |     160    |      1.45      |      120.83     |
> > | **IBCD (Ours)**    |      32.95     |      1     |      0.012     |        1        |
> >
> > **Q5. Can you comment on the choice of the distance function and how it affects the results? For example, some methods employ distance instead of LPIPS in consistency loss [4].**
> >
> > Following your suggestion, we evaluated several commonly used distance functions for the consistency loss, including LPIPS, L1, L2, and pseudo-Huber. Due to time constraints, we conducted these comparisons on the Cat↔Dog 64×64 task for 50K iterations. All distance functions produced similar levels of source faithfulness, but our original choice, LPIPS, yielded substantially better realism, achieving a noticeably lower FID score. These results further support our design choice of using LPIPS for the consistency loss.
> >
> > *Table E. Ablation of consistency loss distance functions on the unpaired cat→dog translation task.*
> > | Distance Function |  FID↓ | PSNR↑ |  SSIM↑ |
> > |-------------------|:-----:|:-----:|:------:|
> > | LPIPS      | 49.13 | 19.70 | 0.6897 |
> > | L1                | 58.77 | 19.74 | 0.6974 |
> > | L2                | 67.82 | 19.92 | 0.7093 |
> > | Pseudo-Huber      | 59.42 | 20.02 | 0.7060 |
> >
> > *Table F. Ablation of consistency loss distance functions on the unpaired dog→cat translation task.*
> > | Distance Function |  FID↓ | PSNR↑ |  SSIM↑ |
> > |:-----------------:|:-----:|:-----:|:------:|
> > | LPIPS     | 27.59 | 19.30 | 0.6660 |
> > | L1                | 37.89 | 19.38 | 0.6818 |
> > | L2                | 47.95 | 19.36 | 0.6864 |
> > | Pseudo-Huber      | 39.57 | 19.32 | 0.6802 |
> >
> > **Q6. Because prior works also incorporated DMD loss, it may be worth briefly noting how your implementation relates to that in [5].**
> >
> > Thank you for pointing out this relevant concurrent work. We have now cited [3] in the revised manuscript.
> >
> > Unlike [3], which uses the DMD loss as the primary training objective in an unpaired setting, their formulation does not guarantee that the source and target samples correspond. As a result, they must introduce an additional L2 constraint between input and output and tune its weight via grid search. The performance reported in [3] is highly sensitive to this weight, and different λ values lead to noticeably different translation behaviors.
> >
> > In contrast, our framework does not rely on such constraints. The DDIB teacher provides valid cross-domain correspondences, allowing IBCD to perform translation without requiring input–output L2 regularization. The DMCD loss in our method is used only as an auxiliary objective to further align the student distribution with the teacher. Moreover, because we adopt the adaptive DMCD formulation, the loss weight self-adjusts during training rather than requiring manual tuning. This design makes our approach more stable and less sensitive to hyperparameter selection compared to [3].
> >
> > [6] Naeem et al. Reliable fidelity and diversity metrics for generative models. ICML, 2020.

---

### Official Review · Reviewer_Dtkz · 2025-10-29

**Soundness:** 3
**Presentation:** 3
**Contribution:** 2
**Rating:** 4
**Confidence:** 4

**Summary:**

The authors propose a distillation approach for the previously introduced method for domain translation using unpaired data, known as the Dual Diffusion Implicit Bridge (DDIB). DDIB utilizes two pre-trained diffusion models for each domain, or one conditional diffusion model pre-trained on multiple domains to “concatenate” PF-ODE between the first domain and the noise distribution, with PF-ODE between the noise distribution and the second domain. This approach enables the creation of a PF-ODE that follows from the first domain through the noise distribution to the second domain. The method Implicit Bridge Consistency Distillation (IBCD) proposed by the authors is, in essence, a combination of consistency distillation and distribution matching distillation, adapted to distill “concatenated” PF-ODE in both directions of the first and second domains with the addition of CycleGAN loss. The authors evaluate their method on toy data as well as unpaired image translation on AFHQ and Celeba-HQ datasets, and compare with previous works on the unpaired image-to-image translation method, including the OpenAI foundational model.

**Strengths:**

- The authors propose the adaptation of a combination of consistency and DMD distillation for DDIB.
- A wide list of competitor methods is considered, including the OpenAI foundational model.
- The authors show that the developed model outperforms competitors on quality of generation.
- A user study is provided.

**Weaknesses:**

- The proposed approach is more like an engineering combination of previously proposed distillation methods to the previously proposed method for unpaired domain translation.
- While the authors show superiority of their method compared to other competitors in the quality of generation, this result is mainly due to the usage of the strong teacher model, which also outperforms competitors. However, there is no comparison of the parameter count and training time of these models. There is a possibility that the teacher model outperforms other baselines due to the usage of significantly more parameters and compute, and the distilled version does the same for the same reason.
- I suspect that in the comparison of inference time in Table 7, usage of different batches may not be correct, since there may be a constant overhead for each call of the neural network; hence, methods that use higher batch sizes may be better in img/sec ratio. Since the key part of all methods is to make some number of forward passes of the neural network, a comparison of 1 forward pass of each network with the same batch of images multiplied by NFE would better demonstrate the computational complexity of each model.
- Some other baselines based on SB theory are missed, such as, but not limited to [1, 2].

[1] Shi Y. et al. Diffusion schrödinger bridge matching //Advances in Neural Information Processing Systems. – 2023. – Т. 36. – С. 62183-62223.

[2] Mokrov P. et al. Energy-guided Entropic Neural Optimal Transport //The Twelfth International Conference on Learning Representations.

**Questions:**

Could the authors provide more details on the number of parameters of all methods?

---

> ### Author Response · Authors · 2025-11-24
>
> **W1. The proposed approach is more like an engineering combination of previously proposed distillation methods to the previously proposed method for unpaired domain translation.**
>
> We emphasize that our framework introduces a novel image translation model that satisfies four critical properties—one-step, unpaired, bidirectional, and non-discriminator—a combination not previously achieved. Importantly, realizing such a framework requires addressing several nontrivial technical challenges. This is not a simple application of existing distillation techniques to a noise-to-image generation setup, because our model is built on diffusion bridges, which involve fundamentally different trajectory structures.
>
> Our contributions include: adapting consistency distillation to diffusion bridge trajectories through custom timestep schedules and training schemes; designing appropriate boundary conditions to enable stable bidirectional training; and parameterizing the model to support bidirectional diffusion paths (Appendix B). Additionally, we introduce auxiliary losses—DMCD, adaptive DMCD, and bidirectional cycle loss—that are tightly integrated with the framework. These losses play a crucial role in improving both realism and fidelity. Extensive ablation studies validate the significance of each component. Accordingly, we believe our method represents a substantive advance rather than an incremental extension.
>
> **W2. The reported quality gains may largely stem from the strong teacher model, which already outperforms the baselines. Without parameter and training-time comparisons, it is unclear whether the teacher and its distilled version excel mainly because they rely on significantly greater capacity and compute.**
>
> To address the concern, we evaluated the generation quality of our EDM teacher checkpoint and the publicly available DDPM checkpoints used by ILVR, EGSDE, and CycleDiffusion. The FID results show that our EDM teacher achieves comparable generation quality to these DDPM models, rather than being substantially better, and it even uses fewer parameters. These observations indicate that our teacher does not introduce an unfair advantage and that the comparison is conducted under similar generation capability.
>
> *Table C. Comparison of generation quality across teacher diffusion models (AFHQ-dog).*
> | Diffusion Model                            | Parameters [M] | Training Steps [M] | FID ↓  |
> |--------------------------------------------|:--------------:|:------------------:|:------:|
> | **EDM (teacher of ours)**                      |      32.95     |         0.8        |  40.91 |
> | DDPM (ILVR, SDEdit, EGSDE, CycleDiffusion) |      93.56     |         1.0        |  41.88 |
>
> For the EDM teacher model, we trained it for 800K steps. The student model was then distilled for an additional 210K steps. It is worth noting that the combined training steps of the teacher and student models (1.1M steps) are comparable to those of the diffusion-based baselines, which were trained for 1M steps, while still incurring over 1000× higher inference cost.
> In addition, to further ensure fairness, we re-implemented EGSDE, SDEdit, and CycleDiffusion using the same EDM backbone and checkpoint, and re-tuned all hyperparameters via grid search to ensure optimal and comparable settings. Across all such controlled comparisons, our method consistently outperforms these baselines under identical backbone and training conditions.  These findings confirm that the gains of our method arise from the proposed training framework itself, rather than differences in the underlying teacher model.
>
> *Table D. Quantitative comparison of unpaired cat→dog translation tasks under the same teacher diffusion model conditions.*
> | Model                | NFE↓   |    FID↓   |   PSNR↑   |   SSIM↑   |
> |----------------------|:-----:|:---------:|:---------:|:---------:|
> | SDEdit (EDM)         | 80    | 59.60     | 17.36     | 0.378     |
> | EGSDE (EDM)          | 80    | 56.84     | 17.77     | 0.370     |
> | CycleDiffusion (EDM) | 160   | 45.97     | 14.43     | 0.558     |
> | **IBCD (ours)**      | **1** | 47.44     | **19.50** | **0.701** |
> | **IBCD† (ours)**     | **1** | **44.77** | **18.04** | **0.663** |

---

> ### Author Response · Authors · 2025-11-24
>
> **W3. Because differing batch sizes create unequal overheads per network call, Table 7’s inference-time comparison may be biased. Comparing a single forward pass with a fixed batch size and scaling by NFE would better capture true computational complexity.**
>
> Following your suggestion, we re-measured the inference time using a fixed batch size of 1 and removed warm-up and other overhead that are not part of the actual forward computation. The updated results are reported in Tab.  7 of the revised manuscript. The results confirm that our method remains much faster than the diffusion baseline. Although the inference time is still slower than GAN-based models, the gap is moderate and considered reasonable given the substantial performance improvement. A summary of the updated Tab. 7  is provided below.
>
> *Table A: Quantitative comparison of model inference times and parameter sizes.*
> | Model          | Parameters [M] |    NFE ↓   | Time [s/img] ↓ | Relative Time ↓ |
> |----------------|:--------------:|:----------:|:--------------:|:---------------:|
> | StarGAN v2     |      64.45     |      1     |     0.0052     |       0.43      |
> | CUT            |      11.39     |      1     |     0.0070     |       0.58      |
> | UNSB           |      14.69     |      5     |      0.077     |       6.42      |
> | ILVR           |      93.56     |    1000    |      13.40     |     1116.67     |
> | SDEdit         |      93.56     |    1000    |      6.78      |      565.00     |
> | EGSDE          |     147.14     |    1000    |      15.89     |     1324.16     |
> | CycleDiffusion |     187.12     | 1000(+100) |      26.03     |     2169.17     |
> | GPT-Image-1    |       >>1      |     >>1    |      30.32     |     2526.67     |
> | DDIB (Teacher) |      32.95     |     160    |      1.45      |      120.83     |
> | **IBCD (Ours)**    |      32.95     |      1     |      0.012     |        1        |
>
> **W4. Some other baselines based on SB theory are missed, such as, but not limited to [1, 2].**
>
> We added five additional baselines involving Schrödinger Bridge and Optimal Transport (Appendix D.7, Tab. 8, and Fig. 11). Our method consistently outperforms all added SB/OT baselines. Moreover, all of the methods fail to scale to our higher-resolution setting with their default configurations (with the exception of Eg-EOT, which operates in the StyleGAN2-ADA latent space), reflecting a curse-of-dimensionality limitation inherent to high-dimensional OT/SB formulations. DSBM also reflects this limitation: its higher source faithfulness results from failing to produce meaningful target-domain translations at higher resolutions, leading to minimal changes and thus significantly worse FID, Density, and Coverage metrics. A summary of the quantitative results on the Cat→Dog task is provided below.
>
> *Table B. Quantitative comparison of unpaired cat→dog translation tasks with OT- and SB-based baselines.*
> | Model     | NFE | Parameters |    FID↓   |   PSNR↑   |   SSIM↑   |  Density↑ | Coverage↑ |
> |-----------|:---:|:----------:|:---------:|:---------:|:---------:|:---------:|:---------:|
> | NOT [3]   |  1  |    9.72M   |   161.54  |   15.12   |   0.566   |   0.531   |   0.072   |
> | DIOTM [5] |  1  |    39.65M  |   75.70   |   12.03   |   0.363   |   1.215   |   0.590   |
> | ASBM [4]  |  4  |   79.58M   |   91.40   |   17.71   |   0.463   |   0.871   |   0.478   |
> | DSBM [1]  | 100 |   131.02M  |   100.08  | **21.24** |   0.532   |   0.750   |   0.396   |
> | EGEOT [2] | 100 |   26.21M   |   53.29   |   15.93   |   0.349   |   1.085   |   0.626   |
> | ours      |  1  |   32.95M   | **47.44** |  _19.50_  | **0.701** | **1.412** | **0.940** |
> | ours†     |  1  |   32.95M   | **44.77** |  _18.04_  | **0.663** | **1.542** | **0.935** |
>
> [3] Korotin, Alexander, Daniil Selikhanovych, and Evgeny Burnaev. "Neural optimal transport." ICLR, 2023.
>
> [4] Gushchin, Nikita, et al. "Adversarial schrödinger bridge matching. "NeuIPS, 2024.
>
> [5] Choi, Jaemoo, et al. "Improving neural optimal transport via displacement interpolation." ICLR, 2025.

---

### Official Review · Reviewer_NQcj · 2025-10-31

**Soundness:** 3
**Presentation:** 2
**Contribution:** 2
**Rating:** 6
**Confidence:** 4

**Summary:**

This paper proposes a novel one-step method called Implicit Bridge Consistency Distillation (IBCD) for unpaired Image-to-Image translation by extending the consistency distillation framework to concatenated source-target PF-ODE obtained via Dual Diffusion Implicit Bridge (DDIB). In addition to this, the authors propose using a Distribution Matching loss with an adaptive reweighing scheme based on the introduced distillation complexity proxy to enhance the realism of samples, and a cycle-consistency loss to improve faithfulness. The proposed approach beats the corresponding baselines across multiple commonly used unpaired image-to-image benchmarks.

**Strengths:**

1. The application of Consistency Distillation to the unpaired image-to-image translation is novel and conceptually interesting.
2. The method proposed in the paper enables simultaneous bidirectional training and one-step inference.
3. The experimental section is extensive, with convincing quantitative and qualitative results, including ablations, failure cases, and a user study.
4. The MRI Contrast Translation experiments suggest potential applicability beyond standard image translation tasks, which strengthens the paper’s general interest.

**Weaknesses:**

1. The current quantitative comparison omits comparison with Optimal Transport methods, such as NOT [1] and/or ASBM [2], which is a significant methodological gap.
2. The two-stage training pipeline raises questions about efficiency and stability. As indicated in Table 4, the training of the first IBCD-only stage consumes the majority of the training time, while the second stage, which enables a better trade-off in the end, accounts for less than 20% of the total training steps. This imbalance suggests that the student initialisation may be suboptimal and that the training dynamics could be unstable when combining objectives.
3. The benefit of adding DMCD and DMCD & Cycle losses to IBCD-only on Figure 3 is not clearly demonstrated. Since DMCD should enhance target distribution realism and cycle consistency should enforce source-target faithfulness, the visual distinctions should be more evident.

References:
- [1] Neural Optimal Transport
- [2] Adversarial Schrodinger Bridge Matching

**Questions:**

1. How does bidirectional training affect performance compared to a unidirectional IBCD model? Does it improve the final trade-off between realism and faithfulness, or does it introduce additional instabilities?
2. What motivates the two-stage training design and the rapid convergence of the second stage? Why is joint training (IBCD + non-adaptive DMCD + Cycle) from the start not feasible?
3. How long does the student model take to converge, and how does its total training time compare to the teacher model’s training time?
4. Could you please provide parameter counts for both teacher and student models, and for the diffusion-based baselines used in the image translation experiments?
5. Could you expand on the MRI Contrast Translation setup? Specifically, how was the IBCD teacher model trained, and did the diffusion-based baselines share the same teacher initialisation?

---

> ### Author Response · Authors · 2025-11-24
>
> **W1. The current quantitative comparison omits comparison with Optimal Transport methods, such as NOT [1] and/or ASBM [2], which is a significant methodological gap.**
>
> We added five additional baselines involving Schrödinger Bridge and Optimal Transport (Appendix D.7, Tab. 8, and Fig. 11). Our method consistently outperforms all added SB/OT baselines. Moreover, all of the methods fail to scale to our higher-resolution setting with their default configurations (with the exception of Eg-EOT, which operates in the StyleGAN2-ADA latent space), reflecting a curse-of-dimensionality limitation inherent to high-dimensional OT/SB formulations. DSBM also reflects this limitation: its higher source faithfulness results from failing to produce meaningful target-domain translations at higher resolutions, leading to minimal changes and thus significantly worse FID, Density, and Coverage metrics. A summary of the quantitative results on the Cat→Dog task is provided below.
>
> *Table B. Quantitative comparison of unpaired cat→dog translation tasks with OT- and SB-based baselines.*
> | Model     | NFE | Parameters |    FID↓   |   PSNR↑   |   SSIM↑   |  Density↑ | Coverage↑ |
> |-----------|:---:|:----------:|:---------:|:---------:|:---------:|:---------:|:---------:|
> | NOT [1]   |  1  |    9.72M   |   161.54  |   15.12   |   0.566   |   0.531   |   0.072   |
> | DIOTM [5] |  1  |    39.65M  |   75.70   |   12.03   |   0.363   |   1.215   |   0.590   |
> | ASBM [2]  |  4  |   79.58M   |   91.40   |   17.71   |   0.463   |   0.871   |   0.478   |
> | DSBM [3]  | 100 |   131.02M  |   100.08  | **21.24** |   0.532   |   0.750   |   0.396   |
> | EGEOT [4] | 100 |   26.21M   |   53.29   |   15.93   |   0.349   |   1.085   |   0.626   |
> | ours      |  1  |   32.95M   | **47.44** |  _19.50_  | **0.701** | **1.412** | **0.940** |
> | ours†     |  1  |   32.95M   | **44.77** |  _18.04_  | **0.663** | **1.542** | **0.935** |
>
> **W2/Q2. What motivates the two-stage training design and the rapid convergence of the second stage? Why is joint training (IBCD + non-adaptive DMCD + Cycle) from the beginning not feasible?**
>
> Our motivation is straightforward: we first need to establish a stable source–target bridge. Early in training, the student’s predictions are still noisy, so introducing DMCD or Cycle loss from the beginning produces gradients that conflict with IBCD, consistently destabilizing optimization and preventing the bridge from forming properly. IBCD is the only loss that reliably constructs this bridge—both in terms of domain mapping and target-distribution alignment—without requiring any source–target pairing heuristics, so we allow the model to fully learn this backbone in Stage 1. Once the bridge is established, Stage 2 performs lightweight yet effective refinement through adaptive DMCD and Cycle loss, which leads to fast convergence because it refines the trajectory already established in Stage 1. In contrast, joint training from the start (IBCD + non-adaptive DMCD + Cycle) disrupts early bridge formation due to conflicting gradients and consistently results in unstable or collapsed optimization. Additionally, auxiliary losses introduce extra computational overhead, so our staged strategy also reduces training cost.
>
> **W3. The benefit of adding DMCD and DMCD & Cycle losses to IBCD-only on Figure 3 is not clearly demonstrated.**
>
> In the IBCD-only case, the basic translation is learned, but several points are incorrectly mapped into the low-density gaps between the rolls. When DMCD is added, these mis-mapped points are pushed toward the roll. The reduction in point density within the inter-roll gaps is clearly observed when looking at the full figure rather than the zoomed-in regions. However, DMCD alone can push too strongly toward high-density areas, causing the mapped manifold to contract and exhibit a smaller volume than the true target manifold.
>
> Adding the cycle loss alleviates this over-concentration effect. When too many points collapse into high-density regions, the model struggles to map them back to the source domain, which increases the cycle-consistency loss and therefore encourages a more balanced expansion of the mapped manifold. Finally, the adaptive DMCD applies DMCD only to points that remain in low-density regions, effectively removing these incorrect mappings without introducing the over-contraction seen with DMCD alone.

---

> ### Author Response · Authors · 2025-11-24
>
> **Q1. How does bidirectional training perform relative to unidirectional IBCD? Does it enhance the realism–faithfulness balance or cause extra instabilities?**
>
> Our main goal is to train a single unified model that supports bidirectional translation. As a byproduct of this design, we can incorporate cycle-consistency loss efficiently without additional memory cost, and this loss plays a crucial role in improving source faithfulness. In short, bidirectional training is central to our objective, and it additionally enables the efficient use of cycle consistency.
>
> We also confirm that bidirectional training does not introduce instability. The training loss curves provided in the revised manuscript (Appendix D.5 and Fig. 10) show stable behavior throughout training. The consistency loss and the cycle loss steadily decrease, and the DMCD loss remains at a stable plateau without noticeable fluctuations. These observations indicate that the proposed bidirectional training scheme remains stable and does not negatively affect the training dynamics.
>
> **Q3. How long does the student converge, and how does its total training time compare with the teacher’s?**
>
> For the EDM teacher model, we trained it for 800K steps (5 days). The student model was then distilled for an additional 210K steps (3 days). It is worth noting that the combined training steps of the teacher and student models (1.1M steps) are comparable to those of the diffusion-based baselines, which were trained for 1M steps, while still incurring over 1000× higher inference cost. Furthermore, the baselines use models whose parameter sizes are about three times larger than ours (see answer to Q4)
>
> **Q4. Could you provide the parameter counts for the teacher, the student, and the diffusion-based baselines used in the image translation experiments?**
>
> We have updated Tab. 7 in the manuscript to include the parameter counts for all baselines. Our teacher and student models share the same architecture, resulting in a parameter size of 32.95M. This is less than one-third of the size of the diffusion-based baselines, while still achieving better performance, which shows that our framework is both lightweight and effective. A summary of the updated Tab. 7  is provided below.
>
> *Table A: Quantitative comparison of model inference times and parameter sizes.*
> | Model          | Parameters [M] |    NFE ↓   | Time [s/img] ↓ | Relative Time ↓ |
> |----------------|:--------------:|:----------:|:--------------:|:---------------:|
> | StarGAN v2     |      64.45     |      1     |     0.0052     |       0.43      |
> | CUT            |      11.39     |      1     |     0.0070     |       0.58      |
> | UNSB           |      14.69     |      5     |      0.077     |       6.42      |
> | ILVR           |      93.56     |    1000    |      13.40     |     1116.67     |
> | SDEdit         |      93.56     |    1000    |      6.78      |      565.00     |
> | EGSDE          |     147.14     |    1000    |      15.89     |     1324.16     |
> | CycleDiffusion |     187.12     | 1000(+100) |      26.03     |     2169.17     |
> | GPT-Image-1    |       >>1      |     >>1    |      30.32     |     2526.67     |
> | DDIB (Teacher) |      32.95     |     160    |      1.45      |      120.83     |
> | **IBCD (Ours)**    |      32.95     |      1     |      0.012     |        1        |
>
> **Q5. Could you elaborate on the MRI Contrast Translation setup? How was the IBCD teacher trained, and did the diffusion baselines share the same teacher initialization?**
>
> We used T1- and T2-weighted brain MRI scans from the BraTS2021 dataset. The teacher model was trained as a two-class conditional diffusion model, with T1 and T2 serving as the conditioning labels. For our method, the teacher was trained using EDM, while the diffusion-based baselines rely on ADM checkpoints. All models have approximately 33M parameters.
>
> We did not reimplement the baselines with an EDM backbone because their architectures and update rules were originally designed for ADM. Re-deriving and re-implementing them under EDM would produce versions that differ substantially from the original methods, which could disadvantage the baselines. To maintain fairness, we used their standard ADM-based implementations.
>
> [3] Shi, Yuyang, et al. "Diffusion schrödinger bridge matching."NeurIPS, 2023.
>
> [4] Mokrov, Petr, et al. "Energy-guided entropic neural optimal transport."ICLR, 2023.
>
> [5] Choi, Jaemoo, et al. "Improving neural optimal transport via displacement interpolation." ICLR, 2025.

---

### Official Review · Reviewer_y4xC · 2025-10-31

**Soundness:** 2
**Presentation:** 2
**Contribution:** 2
**Rating:** 4
**Confidence:** 4

**Summary:**

The paper proposes a novel approach, called IBCD, for solving unpaired image-to-image (I2I) problems. The method suggests training a one-step bidirectional translation map via consistency distillation of the DDIB (Denoising Diffusion Implicit Bridges) trajectory (which consists of the two composed diffusion ODE trajectories: source$\to$ noise and noise $\to$ target). Additionally, the authors propose using DMD loss with adaptive weighting to enhance the realism of the map's outputs and cycle-consistency loss to improve the input-output alignment. The authors empirically validate the importance of the components in the toy 2d experiment and in the unpaired image-to-image translation problems. The method yields superior results compared to the GAN-based and diffusion-based baselines on the AFHQv2 and CelebA-HQ translation benchmarks.

**Strengths:**

1) Combination of the cycle-consistency loss with DMD on the outputs and DDIB distillation is novel;
2) Adaptive weighting of the DMD loss looks promising and demonstrates efficiency in the toy experiment;
3) The method has efficient one-step inference;
4) The method outperforms the GAN-based baselines and most of the diffusion-based baselines (except the teacher DDIB, where IBCD has better alignment but worse FID).

**Weaknesses:**

1) Comparison with the diffusion-based baselines raises questions about fairness. While DDIB teacher and IBCD share the same class-conditional EDM backbone trained by the authors, the results reported in ILVR, EGSDE, and CycleDiffusion are obtained with the discrete-time DDPM backbone introduced in ILVR in 2021. I think unifying the backbone for all the sampling-based diffusion methods is essential for the fair comparison;
2) Several baselines are missing. The authors report quite a comprehensive amount of GAN-based and diffusion-based baselines, but it is essential to perform comparison with optimal transport (OT)-based baselines. GAN-based methods are older and are typically outperformed by diffusion-based methods (thus, I believe, their relevance is limited), while diffusion-based methods often suffer from lower input-output alignment compared to the one-step counterparts (and one of the advantages of IBCD is better alignment compared to e.g. DDIB teacher model). I appreciate adding UNSB, but comparing IBCD to such methods as e.g. NOT [1] and DIOTM [2] would greatly benefit the paper in terms of positioning against one-step baselines;
3) An important related work [3], which proposes to modify the DMD procedure for image-to-image scenarios, is missing;
4) The method description sometimes seems overloaded and overcomplicated (e.g. Equations 9, 10, 11);
5) The method would greatly benefit from studying higher-dimensional problems or a more diverse set of problems e.g. class- or prompt-conditional I2I translation, or translation between different types of domains.

[1] Neural Optimal Transport

[2] Improving Neural Optimal Transport via Displacement Interpolation

[3] Regularized Distribution Matching Distillation for One-step Unpaired Image-to-Image Translation

**Questions:**

In Table 3, the authors present the effect of different components of the method on the performance in terms of FID and PSNR. The improvement in the input-output alignment is expected after adding cycle consistency loss. However, adaptive DMCD strategy seems to be designed for enhancing realism of samples. Could you please tell why it has a pronounced effect on alignment while slightly harming realism?

---

> ### Author Response · Authors · 2025-11-24
>
> **W1. The comparison is not fully fair because the proposed method uses an EDM teacher while the diffusion-based baselines rely on a different backbone (DDPM).**
>
> Thanks for your concrete comments. To address this concern, we evaluated the generation quality of our EDM teacher checkpoint and the publicly available DDPM checkpoints used by ILVR, EGSDE, and CycleDiffusion. The FID results show that our EDM teacher achieves comparable generation quality to these DDPM models, rather than being substantially better, and it even uses fewer parameters. These observations indicate that our teacher does not introduce an unfair advantage and that the comparison is conducted under similar generation capability.
>
> *Table C. Comparison of generation quality across base diffusion models (AFHQ-dog).*
> | Diffusion Model                            | Parameters [M] | Training Steps [M] | FID ↓  |
> |--------------------------------------------|:--------------:|:------------------:|:------:|
> | **EDM (teacher of ours)**                      |      32.95     |         0.8        |  40.91 |
> | DDPM (ILVR, SDEdit, EGSDE, CycleDiffusion) |      93.56     |         1.0        |  41.88 |
>
> In addition, to further ensure fairness, we re-implemented EGSDE, SDEdit, and CycleDiffusion using the same EDM backbone and checkpoint, and re-tuned all hyperparameters via grid search to ensure optimal and comparable settings. Across all such controlled comparisons, our method consistently outperforms these baselines under identical backbone and training conditions.  These findings confirm that the gains of our method arise from the proposed training framework itself, rather than differences in the underlying teacher model.
>
> *Table D. Quantitative comparison of unpaired cat→dog translation tasks under the same teacher diffusion model conditions.*
> | Model                | NFE↓   |    FID↓   |   PSNR↑   |   SSIM↑   |
> |----------------------|:-----:|:---------:|:---------:|:---------:|
> | SDEdit (EDM)         | 80    | 59.60     | 17.36     | 0.378     |
> | EGSDE (EDM)          | 80    | 56.84     | 17.77     | 0.370     |
> | CycleDiffusion (EDM) | 160   | 45.97     | 14.43     | 0.558     |
> | **IBCD (ours)**      | **1** | 47.44     | **19.50** | **0.701** |
> | **IBCD† (ours)**     | **1** | **44.77** | **18.04** | **0.663** |
>
> **W2. Important OT-based baselines are missing. Including one-step methods such as NOT [1] and DIOTM [2]—in addition to UNSB—would significantly strengthen the paper’s positioning.**
>
> We added five additional baselines involving Schrödinger Bridge and Optimal Transport (Appendix D.7, Tab. 8, and Fig. 11), excluding DIOTM [2] without a public implementation. Our method consistently outperforms all added SB/OT baselines. Moreover, all of the methods fail to scale to our higher-resolution setting with their default configurations (with the exception of Eg-EOT, which operates in the StyleGAN2-ADA latent space), reflecting a curse-of-dimensionality limitation inherent to high-dimensional OT/SB formulations. DSBM also reflects this limitation: its higher source faithfulness results from failing to produce meaningful target-domain translations at higher resolutions, leading to minimal changes and thus significantly worse FID, Density, and Coverage metrics. A summary of the quantitative results on the Cat→Dog task is provided below.
>
> *Table B. Quantitative comparison of unpaired cat→dog translation tasks with OT- and SB-based baselines.*
> | Model     | NFE | Parameters |    FID↓   |   PSNR↑   |   SSIM↑   |  Density↑ | Coverage↑ |
> |-----------|:---:|:----------:|:---------:|:---------:|:---------:|:---------:|:---------:|
> | NOT [1]   |  1  |    9.72M   |   161.54  |   15.12   |   0.566   |   0.531   |   0.072   |
> | DIOTM [7] |  1  |    39.65M  |   75.70   |   12.03   |   0.363   |   1.215   |   0.590   |
> | ASBM [4]  |  4  |   79.58M   |   91.40   |   17.71   |   0.463   |   0.871   |   0.478   |
> | DSBM [5]  | 100 |   131.02M  |   100.08  | **21.24** |   0.532   |   0.750   |   0.396   |
> | EGEOT [6] | 100 |   26.21M   |   53.29   |   15.93   |   0.349   |   1.085   |   0.626   |
> | ours      |  1  |   32.95M   | **47.44** |  _19.50_  | **0.701** | **1.412** | **0.940** |
> | ours†     |  1  |   32.95M   | **44.77** |  _18.04_  | **0.663** | **1.542** | **0.935** |

---

> ### Author Response · Authors · 2025-11-24
>
> **W3. The paper does not reference an important related work [3] that adapts the DMD procedure for image-to-image scenarios.**
>
> Thank you for pointing out this relevant concurrent work. We have now cited [3] in the revised manuscript.
>
> Unlike [3], which uses the DMD loss as the primary training objective in an unpaired setting, their formulation does not guarantee that the source and target samples correspond. As a result, they must introduce an additional L2 constraint between input and output and tune its weight via grid search. The performance reported in [3] is highly sensitive to this weight, and different λ values lead to noticeably different translation behaviors.
>
> In contrast, our framework does not rely on such constraints. The DDIB teacher provides valid cross-domain correspondences, allowing IBCD to perform translation without requiring input–output L2 regularization. The DMCD loss in our method is used only as an auxiliary objective to further align the student distribution with the teacher. Moreover, because we adopt the adaptive DMCD formulation, the loss weight self-adjusts during training rather than requiring manual tuning. This design makes our approach more stable and less sensitive to hyperparameter selection compared to [3].
>
> **W4. The method description sometimes seems overloaded and overcomplicated (e.g., Eq. 9, 10, 11).**
>
> Following your suggestion, we have simplified the notation to improve readability. In addition to these revisions, we hope that Algorithms 1,2 also help clarify the procedure.
>
> **W5. The method would greatly benefit from exploring higher-dimensional or more diverse tasks, such as class- or prompt-conditional I2I translation or cross-domain translation.**
>
> Thank you for suggesting these promising future research directions. Extending the method to higher-resolution settings could benefit from leveraging latent-space diffusion, which has become a common strategy in recent diffusion-model literature. Studying multi-class or prompt-conditional image-to-image translation is also an exciting direction. Our framework is already built on a class-conditional model and relies on target-domain condition inputs, which naturally enables extensions to class- or prompt-conditional I2I translation. While these directions are highly interesting, they represent substantial undertakings on their own, so we consider them beyond the scope of this work and leave them for future investigation.
>
> **Q1. Could you explain why adaptive DMCD greatly improves alignment but slightly harms realism, even though it seems to be designed to improve realism?**
>
> The behavior you observed is expected because adaptive DMCD is designed to moderate the overly aggressive effect of vanilla DMCD. While vanilla DMCD improves realism by pushing the translated distribution closer to the target data distribution, it can also increase divergence from the teacher model and therefore reduce faithfulness to the source domain.
>
> Adaptive DMCD resolves this by adjusting the strength of the DMCD loss according to the difficulty of each PF-ODE trajectory. It applies stronger DMCD supervision to trajectories that the student struggles to translate accurately, especially those near the decision boundary. In contrast, trajectories that are already well aligned and can be handled sufficiently with only the consistency loss receive a weaker DMCD signal. This avoids unnecessary distortion of regions that the student already models well.
>
> As a result, adaptive DMCD shows a slight reduction in realism compared to vanilla DMCD but achieves a clear improvement in source faithfulness, which is the intended trade-off.
>
> [4] Gushchin, Nikita, et al. "Adversarial schrödinger bridge matching. "NeuIPS, 2024.
>
> [5] Shi, Yuyang, et al. "Diffusion schrödinger bridge matching."NeurIPS, 2023.
>
> [6] Mokrov, Petr, et al. "Energy-guided entropic neural optimal transport."ICLR, 2023.
>
> [7] Choi, Jaemoo, et al. "Improving neural optimal transport via displacement interpolation." ICLR, 2025.

---

> > ### Comment · Reviewer_y4xC · 2025-11-27
> >
> > Thanks for your rebuttal and providing new experiments. However, some of my concerns regarding the comparisons remain unaddressed. The diffusion-based baselines with EDM backbone still lack CycleDiffusion, which is an important two-sided baseline that typically has better faithfulness than the one-sided SDEdit and EGSDE.
> >
> > Besides that, my main remaining concern is the improper comparison with OT-based methods. In short, the comparison needs to include at least one adversarial-based (one-step, NOT-like) OT method with relatively recent architecture to be considered fair.
> >
> > While I appreciate adding Schrödinger Bridge-based methods, they are indeed typically hard to scale and may produce less realistic outputs. At the same time, one-step methods involving adversarial training for realism and some distance function for alignment, typically produce better results and suffer less from such scaling limitations. I think adding only NOT with the corresponding $<10M$ parameters architecture (much weaker than the IBCD's) does not properly reflect the current state of one-step OT methods. Such papers as UOTM [1] or DIOTM [2] demonstrate that with the DDPM++/NCSN++ architectures adversarial OT methods (including NOT) are capable of performing well even in higher dimensions. I would suggest the authors to include at least one OT-based method from this family with this type of architecture (comparable to the one of IBCD). Running IBCD on one of the baselines' benchmarks would also be beneficial, since it would remove the factor of possibly non-suitable default configuration and remove the need in running the corresponding code. In case of DIOTM, to my knowledge, its code is publicly available at OpenReview.
> >
> > For now, due to the aforementioned limitations in comparisons, I keep my original score.
> >
> > [1] Generative Modeling through the Semi-dual Formulation of Unbalanced Optimal Transport
> >
> > [2] Improving Neural Optimal Transport via Displacement Interpolation

---

> ### Author Response · Authors · 2025-11-30
>
> Thank you for pointing out the missing components in our previous response and for kindly locating the source code for the DIOTM baseline. We are pleased to note that your comment allowed us to further improve the completeness of our comparisons.
>
> 1. As requested, we additionally added the results for **CycleDiffusion with the EDM backbone**, and updated them in Table D. While EDM-CycleDiffusion achieves reasonably good FID scores, it still underperforms our method in FID and, more importantly, exhibits significantly lower source faithfulness.
>
> 2. Additionally, we conducted experiments with the **DIOTM baseline** and updated the results in Table B, Table 8, and Figure 11. DIOTM uses a model architecture and parameter count that are highly comparable to ours and demonstrates substantially better FID than NOT. Nevertheless, under our high-resolution settings, DIOTM falls short of our method in FID and shows noticeably lower source faithfulness.
>
> We have reflected these updates in our previously submitted responses as well as in our replies to the other reviewers. For clarity, we restate the updated results from the two revised tables below. We hope that these updates address your concerns regarding the fairness of the comparisons.
>
> *Table D. Quantitative comparison of unpaired cat→dog translation tasks under the same teacher diffusion model conditions.*
> | Model                | NFE↓   |    FID↓   |   PSNR↑   |   SSIM↑   |
> |----------------------|:-----:|:---------:|:---------:|:---------:|
> | SDEdit (EDM)         | 80    | 59.60     | 17.36     | 0.378     |
> | EGSDE (EDM)          | 80    | 56.84     | 17.77     | 0.370     |
> | CycleDiffusion (EDM) | 160   | 45.97     | 14.43     | 0.558     |
> | **IBCD (ours)**      | **1** | 47.44     | **19.50** | **0.701** |
> | **IBCD† (ours)**     | **1** | **44.77** | **18.04** | **0.663** |
>
>
> *Table B. Quantitative comparison of unpaired cat→dog translation tasks with OT- and SB-based baselines.*
> | Model     | NFE | Parameters |    FID↓   |   PSNR↑   |   SSIM↑   |  Density↑ | Coverage↑ |
> |-----------|:---:|:----------:|:---------:|:---------:|:---------:|:---------:|:---------:|
> | NOT [1]   |  1  |    9.72M   |   161.54  |   15.12   |   0.566   |   0.531   |   0.072   |
> | DIOTM [7] |  1  |    39.65M  |   75.70   |   12.03   |   0.363   |   1.215   |   0.590   |
> | ASBM [4]  |  4  |   79.58M   |   91.40   |   17.71   |   0.463   |   0.871   |   0.478   |
> | DSBM [5]  | 100 |   131.02M  |   100.08  | **21.24** |   0.532   |   0.750   |   0.396   |
> | EGEOT [6] | 100 |   26.21M   |   53.29   |   15.93   |   0.349   |   1.085   |   0.626   |
> | ours      |  1  |   32.95M   | **47.44** |  _19.50_  | **0.701** | **1.412** | **0.940** |
> | ours†     |  1  |   32.95M   | **44.77** |  _18.04_  | **0.663** | **1.542** | **0.935** |

---

### Author Response · Authors · 2025-11-24
**General Response**

We thank all reviewers for their constructive and thoughtful feedback.

Several reviewers recognized our method as novel and conceptually promising (‘y4xC’, ‘NQcj’, ‘fmc9’), acknowledged the validation of key components through toy experiments (‘y4xC’, ‘Dtkz’), and appreciated the breadth of our evaluation, including user studies and GPT-based comparisons (‘NQcj’, ‘Dtkz’, ‘fmc9’).

Importantly, all reviewers (‘y4xC’, ‘NQcj’, ‘Dtkz’, ‘fmc9’) also acknowledged that one-step bidirectional image translation is feasible within our framework, which we appreciate as a key recognition of the contribution.
Below, we summarize the main concerns and how we addressed them, followed by detailed point-by-point responses. Updates made in the main manuscript and the appendix are highlighted in blue.

## Key Updates in This Rebuttal

**1. Clarification on Parameter Size and Inference Time (‘y4xC’, ‘NQcj’, ‘Dtkz’, ‘fmc9’)**

We updated Tab. 7 to report the parameter size and inference latency of all baselines under identical conditions (batch size = 1, measuring only the neural network forward time). Importantly, our model uses even fewer parameters than diffusion-based baselines, ensuring that the observed performance gains do not stem from model capacity but directly from the proposed algorithmic contribution. A summary of Tab. 7 is provided below.

*Table A: Quantitative comparison of model inference times and parameter sizes.*
| Model          | Parameters [M] |    NFE ↓   | Time [s/img] ↓ | Relative Time ↓ |
|----------------|:--------------:|:----------:|:--------------:|:---------------:|
| StarGAN v2     |      64.45     |      1     |     0.0052     |       0.43      |
| CUT            |      11.39     |      1     |     0.0070     |       0.58      |
| UNSB           |      14.69     |      5     |      0.077     |       6.42      |
| ILVR           |      93.56     |    1000    |      13.40     |     1116.67     |
| SDEdit         |      93.56     |    1000    |      6.78      |      565.00     |
| EGSDE          |     147.14     |    1000    |      15.89     |     1324.16     |
| CycleDiffusion |     187.12     | 1000(+100) |      26.03     |     2169.17     |
| GPT-Image-1    |       >>1      |     >>1    |      30.32     |     2526.67     |
| DDIB (Teacher) |      32.95     |     160    |      1.45      |      120.83     |
| **IBCD (Ours)**    |      32.95     |      1     |      0.012     |        1        |

**2. Expanded Comparison with SB- and OT-based Methods (‘y4xC’, ‘NQcj’, ‘Dtkz’)**

We added all five baselines requested by the reviewers, which involve Schrödinger Bridge and Optimal Transport (Appendix D.7, Table 8, and Figure 11). Our method consistently outperforms all added SB/OT baselines. Moreover, all of the methods fail to scale to our higher-resolution setting with their default configurations (with the exception of Eg-EOT, which operates in the StyleGAN2-ADA latent space), reflecting a curse-of-dimensionality limitation inherent to high-dimensional OT/SB formulations. DSBM also reflects this limitation: its higher source faithfulness results from failing to produce meaningful target-domain translations at higher resolutions, leading to minimal changes and thus significantly worse FID, Density, and Coverage metrics.  A summary of the quantitative results on the Cat→Dog task is provided below.

*Table B. Quantitative comparison of unpaired cat→dog translation tasks with OT- and SB-based baselines.*
| Model     | NFE | Parameters |    FID↓   |   PSNR↑   |   SSIM↑   |  Density↑ | Coverage↑ |
|-----------|:---:|:----------:|:---------:|:---------:|:---------:|:---------:|:---------:|
| NOT       |  1  |    9.72M   |   161.54  |   15.12   |   0.566   |   0.531   |   0.072   |
| DIOTM     |  1  |    39.65M  |   75.70   |   12.03   |   0.363   |   1.215   |   0.590   |
| ASBM      |  4  |   79.58M   |   91.40   |   17.71   |   0.463   |   0.871   |   0.478   |
| DSBM      | 100 |   131.02M  |   100.08  | **21.24** |   0.532   |   0.750   |   0.396   |
| EGEOT     | 100 |   26.21M   |   53.29   |   15.93   |   0.349   |   1.085   |   0.626   |
| ours      |  1  |   32.95M   | **47.44** |  _19.50_  | **0.701** | **1.412** | **0.940** |
| ours†     |  1  |   32.95M   | **44.77** |  _18.04_  | **0.663** | **1.542** | **0.935** |

**3. Training Stability and Visualization (‘NQcj’, ‘fmc9’)**

Several reviewers requested clarification on two-stage training stability and convergence. In response, we included additional analyses and visualizations demonstrating stable optimization behavior and clearer training dynamics (Appendix D.5 and Fig. 10). Aside from the loss perturbation at 20k due to the introduction of DMCD and cycle loss, the results clearly demonstrate the training stability.

---

### Meta-Review · Area_Chair_K6wy · 2026-01-07

**Summary:**

The paper presents a method for bidirectional unpaired image-to-image translation based on several carefully designed optimization strategies. It received mixed ratings from four reviewers. The major concerns from them include missing discussion on and comparison with several important, closely related existing works, the combination nature of the designed overall model, and the lack of key ablation studies to show the effectiveness of different optimization strategies, and the model inference efficiency. The provided rebuttal did not fully address the concerns raised by the reviewers. Great efforts are still needed to fully handle those issues. Based on the overall score distribution and comments from the reviewers, AC finally decided to recommend a rejection of this submission this time.

**Reviewer Concerns:**

The major concerns from the reviewers include the following aspects: (i) missing discussion on and comparison with several important, closely related existing works, (ii) the combination nature of the designed overall model, and (iii) the lack of key ablation studies to show the effectiveness of different optimization strategies, and the model inference efficiency. AC feels that significant efforts are still necessary to address all of these issues.

**Reviewer Scores:**

The original score distribution is 4, 6, 4, 6. The reviewers did not show intentions to further raise the scores.

---

### Decision · Program_Chairs · 2026-01-26

Reject